# Zero-shot Active Mapping via Fused 360-BEV Representations and Vision–Language Models

**Yuanze Wang** [* 1]  **Dianxi Shi** [* 2]  **Yuetian Wang** [1]  **Shiming Song** [2]  **Haikuo Peng** [3]  **Chunping Qiu** [2]
**Mengzhu Wang** [4]

## Abstract

Active mapping enables embodied agents to understand and interact in previously unseen environments. However, most methods struggle to achieve zero-shot generalization to large-scale scenes and lack support for language instructions. We propose a VLM-based active mapping method that achieves zero-shot mapping while facilitating language-driven human–agent interaction. First, we introduce a 360-BEV representation that integrates omnidirectional semantics with BEV-aligned geometric structure to enhance scene understanding. Second, we develop a candidate waypoint generation strategy that allows the VLM-driven agent to select informative 2D waypoints in image space and back-project them into executable metric actions in 3D space, enabling the VLM to plan in its strongest modality. Third, we design a VLM-based depth-first exploration agent that decomposes the scenes into explorable regions, selects informative waypoints within each region, and organizes them into a topological tree. The agent follows the depth-first exploration policy to achieve thorough coverage of large-scale scenes. Without task-specific training, our method outperforms the strongest baseline, improving coverage and AUC by approximately 13.25% and 14.00%, respectively, while enabling language-conditioned interaction.

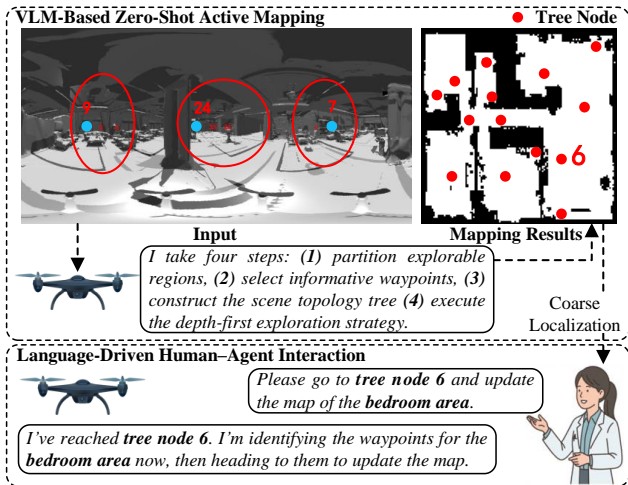

*Figure 1.* We propose a VLM-based active mapping method that achieves zero-shot mapping while facilitating language-driven human–agent interaction.

## 1. Introduction

Active mapping is a fundamental capability for embodied agents to understand previously unseen environments, supporting a broad range of applications such as household robotics (Jung et al., 2023; Man et al., 2024) and autonomous driving (Liu et al., 2023; 2024). While classical active mapping methods (Yan et al., 2023; Placed et al., 2023; Chen et al., 2025a) have demonstrated efficacy in small-scale or simplistic scenes, their generalization to large-scale and complex scenes remains under-explored. Recent work, GLEAM (Chen et al., 2025b), has made significant progress by introducing a large-scale training dataset and a tailored reinforcement learning strategy, improving performance in large-scale scenes and establishing a challenging benchmark. However, GLEAM with small model capacity exhibits limited zero-shot generalization, and may require additional fine-tuning when encountering environments whose style differs significantly from the training set. Moreover, existing active mapping methods are not designed to follow natural-language instructions (Chaplot et al., 2020; Sun et al., 2025; Peralta et al., 2020), limiting human–agent interaction in practical scenarios (e.g., updat-

---
[*]Equal contribution [1]MoE Key Lab of Artificial Intelligence, AI Institute, Shanghai Jiao Tong University, Shanghai, China [2]Intelligent Game and Decision Lab (IGDL), Beijing, China [3]College of Computer Science and Technology, National University of Defense Technology, Changsha, China [4]Hebei University of Technology, Tianjin, China. Correspondence to: Dianxi Shi <dxshi@nudt.edu.cn>.

ing or reconstructing user-specified regions).

Recently, vision–language models (VLMs) (Hurst et al., 2024; Bai et al., 2025; Zhang et al., 2024) have demonstrated strong zero-shot capabilities in cross-modal scene understanding and reasoning, and naturally support language-driven interaction. These properties highlight a promising opportunity to build active mapping methods that support both zero-shot generalization and instruction following. However, a key obstacle is that VLMs without task-specific fine-tuning are ill-suited to directly produce 3D metric actions. Instead, they are most effective when making decisions in image spaces. Therefore, an efficient representation is necessary to unlock the full potential of VLMs for active mapping. Directly adopting scene representation used in classical methods is suboptimal: (i) pinhole images (Kwon et al., 2023; Lluvia et al., 2021) lack omnidirectional semantics and geometry due to their limited field of view; and (ii) navigation-height BEV maps (Chen et al., 2025b) capture scene geometry but often lack rich semantics and can be severely affected by occlusions, preventing omnidirectional scene perception. This motivates the design of an omnidirectional representation that jointly supports semantic and geometric understanding.

In this paper, we propose a VLM-based active mapping method that achieves zero-shot mapping in large-scale scenes while naturally facilitating language-driven human–agent interaction, as shown in Fig. 1. Our method consists of three key components. First, we introduce a fused 360–BEV representation that combines omnidirectional semantic with BEV-aligned geometric structure, enhancing omnidirectional semantic–geometric awareness for VLM. Second, we develop a candidate waypoint generation strategy, enabling the VLM to select informative waypoints in its preferred image space and then back-project the selected 2D waypoints into executable metric actions in the 3D space, while better aligning semantics and geometry in the 360-BEV representation. Third, we design a VLM-based depth-first exploration agent that decomposes scenes into distinct explorable regions, selects informative waypoints within each region, and organizes them into a scene topological tree. A global BEV map maintains the scene topology tree as memory, and the agent follows the depth-first exploration policy, backtracking to the nearest unexplored node whenever no unexplored areas remain in the current branch.

Extensive experiments demonstrate that our zero-shot method outperforms the strongest baseline that relies on extensive training, improving coverage and AUC by approximately 13.25% and 14.00%, respectively. Additionally, our method naturally supports language-driven human–agent interaction, such as updating maps of regions specified by language instructions. In summary, our primary contributions are: (1) We propose a fused 360-BEV representation that improves omnidirectional semantic–geometric perception in VLM. (2) We design a candidate waypoint generation strategy to unlock the potential of VLMs for active mapping while better aligning semantics and geometry in the 360-BEV representation. (3) We introduce a VLM-based depth-first exploration agent that enables zero-shot active mapping while naturally facilitating language-driven human–agent interaction. These designs show that a well-crafted representation and decision framework can unleash the potential of VLMs for embodied active mapping.

## 2. Related Work

**Active Mapping.** Active mapping is a core capability for embodied agents to understand and interact in novel environments. Active SLAM methods (Chen et al., 2019; Sun et al., 2020; Carlone et al., 2014) jointly address localization and mapping. In contrast, exploration-centric methods (Chen et al., 2019; Georgakis et al., 2022) focus on exploration and mapping while assuming poses are available. Many of these methods optimize both scene coverage and high-fidelity 3D appearance, which imposes substantial storage and compute costs. Consequently, they are often evaluated in small, in-distribution environments with few rooms and struggle to generalize to large, cluttered real-world settings (Chang et al., 2017; Straub et al., 2019; Yeshwanth et al., 2023; Wang et al., 2024). To scale active mapping to more complex scenes, GLEAM (Chen et al., 2025b) introduces a large-scale benchmark, where reported baselines exhibit limited zero-shot generalization. To reduce storage and computation in large-scale settings, GLEAM emphasizes exploration and coverage by constructing textureless geometric maps that retain structural semantics while preserving privacy (Zhou et al., 2022). Following this setting, we focus on exploration and coverage, introducing a VLM-driven pipeline that enables zero-shot generalization while facilitating human–agent interaction.

**Representations.** Occupancy-based methods (Ramakrishnan et al., 2020) predict free and occupied space to support efficient exploration and navigation in 3D environments. Other work (Pan et al., 2024; Jiang et al., 2024; Pan et al., 2022) uses neural implicit fields for high-fidelity active mapping. To mitigate slow mapping with neural implicit fields, NARUTO (Feng et al., 2024) proposes a hybrid representation with hash grids at multiple resolutions for more efficient active 3D mapping. With the advent of 3D Gaussian Splatting, several methods (Xu et al., 2025; Li et al., 2025b; Jin et al., 2025) deliver fast and high-quality rendering and accelerate progress in active mapping. However, these methods rely on pinhole camera inputs with a limited field of view and thus cannot capture omnidirectional scene information. In addition, neural implicit fields remain slow (Lee et al., 2022), and 3D Gaussian Splatting

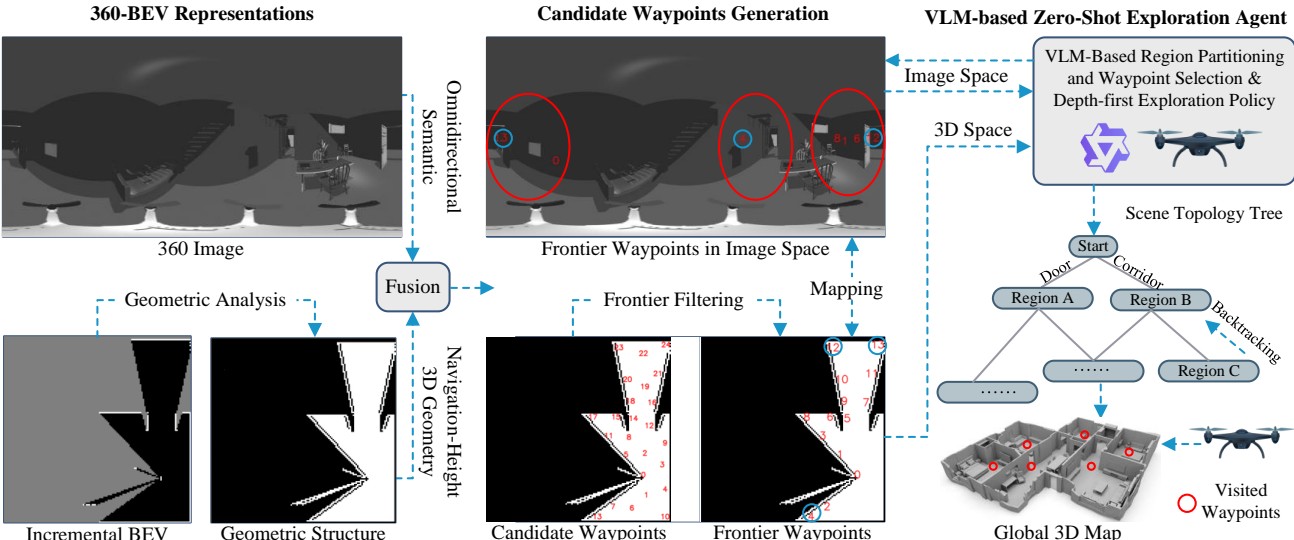

**360-BEV Representations**

360 Image

Omnidirectional Semantic

Geometric Analysis

Fusion

Incremental BEV  Geometric Structure

Navigation-Height 3D Geometry

**Candidate Waypoints Generation**

Frontier Waypoints in Image Space

Frontier Filtering  Mapping

Candidate Waypoints  Frontier Waypoints

**VLM-based Zero-Shot Exploration Agent**

VLM-Based Region Partitioning and Waypoint Selection & Depth-first Exploration Policy

Image Space

3D Space

Scene Topology Tree

Start

Door  Corridor

Region A  Region B

Backtracking

......  Region C

......

Global 3D Map

Visited Waypoints

*Figure 2.* Overview of our pipeline. We construct the fused 360-BEV representation to provide the agent with omnidirectional semantic and geometric perception. To unlock the potential of VLMs for active mapping while aligning semantics and geometry in the 360-BEV, we propose a candidate waypoint generation strategy that allows the VLM to select informative waypoints in its preferred image space while mapping them into executable metric actions in 3D space. Finally, the VLM-based agent partitions the scene into explorable regions and selects waypoints with high information gain for each region. These selected waypoints construct a scene topology tree that guides the agent to execute a depth-first exploration policy, enabling zero-shot generalization and facilitating language-driven interaction.

can be memory- and compute-intensive (Jiang et al., 2025; Jin et al., 2024), which limits scalability to large scenes. To better support large-scale scenes, GLEAM adopts pose history and an efficient egocentric BEV at the navigation height. This representation is susceptible to occlusions that reduce omnidirectional awareness. In contrast, we propose an efficient omnidirectional scene representation that fuses semantics and geometry to improve active mapping.

**Exploration Policies.** Classical heuristic methods (Li et al., 2025a; Ramakrishnan et al., 2020; Chen et al., 2019) struggle to adapt to unseen environments with novel topologies. Information-gain policies (Bircher et al., 2016; Isler et al., 2016) drive active mapping by exploiting scene uncertainty. Recent methods estimate uncertainty with neural implicit fields (Ran et al., 2023; Yan et al., 2023) and with 3D Gaussian Splatting (Chen et al., 2025a; Jiang et al., 2025). Existing exploration policies struggle to achieve zero-shot generalization in unseen large-scale scenes. Recent work (Wu et al., 2026; Wan et al., 2025) further strengthens spatial reasoning in MLLMs. VLMs have recently been introduced into active SLAM (Jiang et al., 2025), but are coupled with memory- and compute-intensive 3DGS representations to ensure high-fidelity 3D maps, limiting their scalability to large-scale environments. Our method focuses on leveraging VLMs to achieve zero-shot active mapping in large-scale scenes, enabling efficient exploration. Unlike uncertainty-driven policies that require explicit modeling scene uncertainty, our agent grounds its decisions in the semantic structure perceived by the VLM, which transfers

across unseen topologies without per-scene tuning.

## 3. Method

An overview of our method is shown in Fig. 2. Following GLEAM, we consider a simulated aerial agent equipped with onboard cameras and an IMU, which incrementally collects observations for active mapping. At each time step $t$, we construct a fused 360–BEV representation to enhance omnidirectional semantic–geometric perception of the agent (Sec. 3.1). Conditioned on this representation, we propose a candidate waypoint generation strategy that allows the VLM to select informative waypoints in its preferred image space while mapping them into executable metric actions in 3D space (Sec. 3.2). To achieve zero-shot active mapping in large-scale scenes, we further propose a VLM-driven depth-first exploration agent, while facilitating language-driven human–agent interaction (Sec. 3.3).

### 3.1. Fused 360-BEV Representations

We construct a 360-BEV representation $O_t$ that enhances the omnidirectional semantic–geometric perception of the VLM-based agent. Omnidirectional semantics are obtained from a 360 image $I_t$ produced via a cubemap projection (Greene, 1986) following GLEAM's rendering pipeline (Chen et al., 2025b). Geometry is provided by the BEV occupancy grid $B_t \in \{-1, 0, 1\}^{H \times W}$ at the robot's navigable height, where -1, 0, and 1 denote known free space, unknown space, and obstacles, respectively. The BEV map

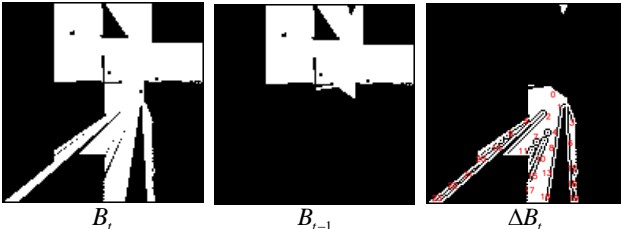

$$B_t \qquad\qquad B_{t-1} \qquad\qquad \Delta B_t$$

*Figure 3.* Illustration of the incremental BEV map and the frontier points derived from it. The frontier points are then projected onto 360 images, enabling the VLM to perform decision-making directly in its preferred image space.

$B_t$ is updated incrementally in the world frame as exploration proceeds, providing the agent with persistent memory, enabling efficient backtracking. As shown in Fig. 3, to focus the agent on the currently visible scene, we use the incremental BEV $\Delta B_t$ at each step:

$$\Delta B_t = B_t \setminus B_{t-1}, \qquad (1)$$

By geometrically analyzing the incremental BEV $\Delta B_t$, we extract the traversable regions $\mathcal{C}_t$ and the frontier edges $\Gamma_t$ to assist the agent in active mapping, where frontier edges are defined as boundaries at which free cells are adjacent to unknown cells. Directly requiring a VLM to interpret BEV geometry and associate it with semantics in the 360 image is inefficient. Intuitively, aligning semantic cues from 360 with geometric structures in the BEV map via anchor points better enables the VLM to jointly reason about semantics and geometry (see Sec. 3.2). Although the explicit data flow is unidirectional (BEV $\rightarrow$ 360), semantic information feeds back to geometry *implicitly* through exploration: the VLM's semantic selection determines where the agent moves, and the resulting depth observations update the BEV—so a room entrance missed by the BEV due to occlusion can still be recovered from 360° semantics in a subsequent step.

### 3.2. Candidate Waypoints Generation

While the agent must plan actions in 3D, general vision–language models are not well-suited to emitting 3D metric actions in embodied environments. To unlock the potential of VLMs for active mapping while better aligning semantics and geometry in the 360-BEV representation, we design a candidate waypoint generation strategy that allows the VLM to select high–information-gain waypoints in its preferred image space. A subsequent geometric lifting step maps the selected 2D waypoint to the executable 3D metric action that directly drives active mapping. Concretely, within the traversable region $\mathcal{C}_t$ extracted from the incremental BEV $\Delta B_t$, we generate $k$ candidate waypoints that are uniformly distributed, edge-safe, and well-separated. We first enforce an edge-safe margin $d_{m}in$ to avoid collisions with obstacles. We then initialize the candidate waypoints

using farthest-point sampling (Eldar et al., 1997) that prioritizes locations far from start point to avoid clustering near the start and to encourage large exploratory moves. We prune candidate waypoints with similar regional coverage using a KD-tree, enforcing a minimum separation radius $r_{min}$. To increase spatial coverage density, we backfill with the farthest feasible points that satisfy the above constraints until the budget $k$ is met. Finally, we obtain the frontier waypoints $\mathcal{W}_t^\Gamma$ filtered by the frontier edges $\Gamma_t$, retaining only those within a radius $d_f$ of $\Gamma_t$ to concentrate exploration near frontiers, as illustrated in Fig. 3:

$$\mathcal{W}_t^\Gamma = \left\{ \mathbf{w} \in \mathcal{W}_t \ \middle| \ \min_{\mathbf{q} \in \Gamma_t} \|\mathbf{w} - \mathbf{q}\|_2 \le d_f \right\}. \qquad (2)$$

This frontier filter is omitted for language-guided human–agent interaction, which may target the entire scene rather than only frontier regions.

The generated candidate waypoints on the BEV map encode key geometric priors, including uniform coverage of traversable space and proximity to exploration frontiers. These waypoints are then projected into the 360 image to align semantics, enabling the VLM to reason jointly about scene semantics and geometry within its preferred image modality. We first convert each candidate waypoint $(x_i, y_i)$ in BEV map to 3D world coordinates using the grid resolution $(\Delta x, \Delta y)$, the BEV origin $(x_{\min}, y_{\min})$, and the navigation height $h_{\text{motion}}$. The resulting point is then mapped to the camera frame via the world-to-camera pose $\mathbf{T}_{w \rightarrow c}$:

$$\begin{bmatrix} x_{c,i} \\ y_{c,i} \\ z_{c,i} \end{bmatrix} = \mathbf{T}_{w \rightarrow c} \begin{bmatrix} x_{w,i} = x_{\min} + \left(x_i + \frac{1}{2}\right)\Delta x \\ y_{w,i} = y_{\min} + \left(y_i + \frac{1}{2}\right)\Delta y \\ z_{w,i} = h_{\text{motion}} \end{bmatrix}, \qquad (3)$$

where the $+\frac{1}{2}$ term centers the waypoint within the grid cell. Finally, we project each candidate waypoint onto the 360 image using the standard equirectangular model to obtain pixel coordinates $(u_i, v_i)$:

$$\begin{aligned} u_i &= \frac{\operatorname{atan2}(x_{c,i}, z_{c,i}) + \pi}{2\pi}\, w, \\ v_i &= \frac{\frac{\pi}{2} - \operatorname{atan2}(-y_{c,i}, \sqrt{x_{c,i}^2 + z_{c,i}^2})}{\pi}\, h, \end{aligned} \qquad (4)$$

where $w, h$ are the width and height of the 360 image. Because the projection exhibits perspective foreshortening in the 360 image, distant waypoints tend to overlap, causing ID collisions and spurious VLM decisions. To mitigate this, we perform non-maximum suppression with radius $r_p$, removing waypoints whose pixel locations lie within $r_p$ of the higher-priority waypoints, where priority is defined by a larger world-space distance from the current pose so as to favor larger motion steps. The candidate waypoints generation pipeline yields a set of 2D–3D correspondences

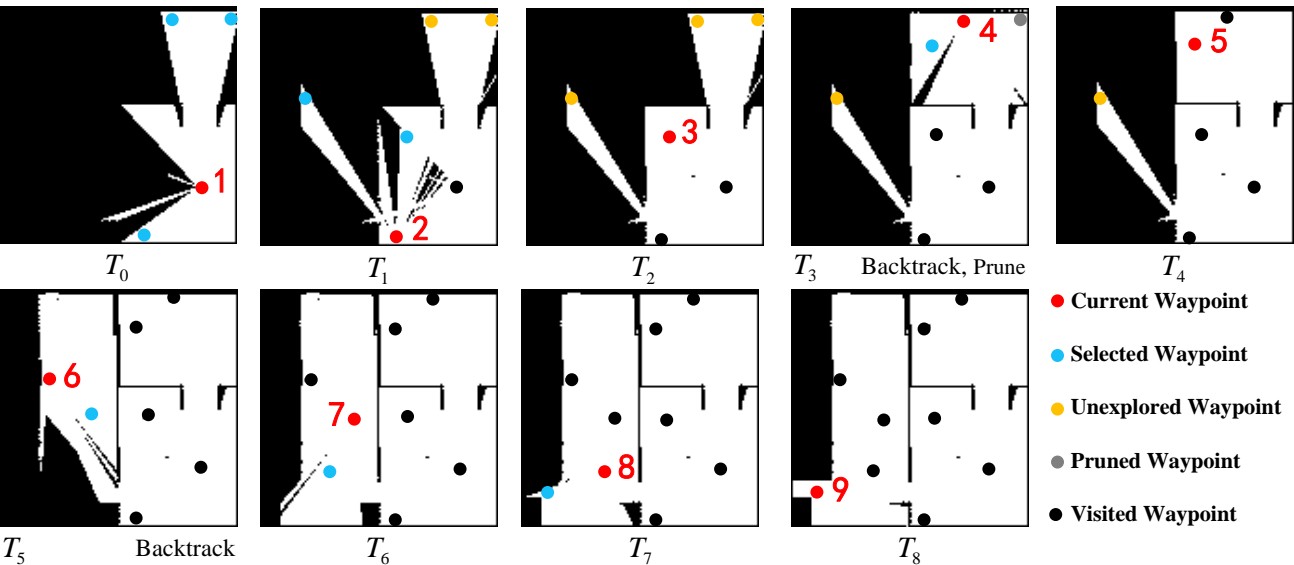

*Figure 4.* Illustration of VLM-based depth-first exploration. The VLM partitions regions and selects waypoints to grow a scene topology tree. At each step, the agent moves to the nearest selected waypoint, while other unexplored waypoints become new nodes in the global BEV memory. As exploration proceeds, candidate waypoints that no longer lie on the frontier are pruned. When the current branch contains no explorable nodes, the agent backtracks to the nearest unexplored node.

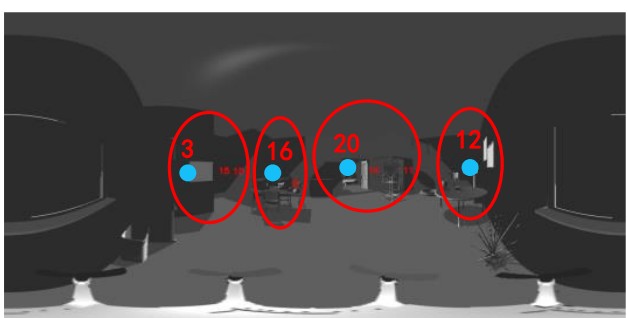

*Figure 5.* Illustration of VLM-driven region partitioning and waypoint selection. Red circles indicate the partitioned regions, and blue circles mark the selected waypoint. The IDs of selected points are enlarged for better visibility.

$\left\{ (u_i, v_i), (x_{w,i}, y_{w,i}, z_{w,i}) \right\}$ in the 360 image and world coordinate frame. Leveraging these correspondences, the VLM selects 2D waypoints in the 360 image, which are then lifted to 3D actions via the 2D–3D correspondences to directly drive active mapping. This design improves decision reliability: the VLM only reasons about discrete image-space candidate waypoints, while geometric validity (collision-free placement and edge-safety) is guaranteed by the candidate generation stage, so the VLM's selections always map to feasible actions, keeping the whole pipeline robust over long-horizon exploration.

### 3.3. VLM-Based Zero-Shot Exploration Agent

To achieve zero-shot generalization in active mapping and support language-driven human-agent interaction, we de-

sign a VLM-based depth-first exploration agent. We first abstract the complex scene as a topological tree whose nodes correspond to explorable regions (e.g., rooms or sub-areas), and whose edges represent connections for explorable regions (e.g., doors and corridors). Concretely, at iteration $t$ the agent acquires a 360 image $I_t$ with a set of candidate waypoints. As shown in Fig. 5, the VLM-based agent decomposes $I_t$ into a set of explorable regions and establishes correspondences between candidate waypoints and these regions.

$$\mathcal{R}_t^{M_t} = f_{\text{VLM}}(I_t), \tag{5}$$

where $M_t$ denotes the number of explorable regions at step $t$. For each region, the VLM then selects the candidate waypoint $w_i^{\star}$ with the highest information gain, which provides access to a larger number of unexplored areas:

$$\{w_i^{\star}\}_{i=1}^{M_t} = f_{\text{VLM}}(\mathcal{R}_t^{M_t}, \mathcal{W}_t^{\Gamma}, I_t). \tag{6}$$

These selected waypoints act as exploration nodes and are used to incrementally construct a scene topology tree $\mathcal{V}_t$, which is maintained in the global BEV map to provide persistent memory for decision making:

$$\mathcal{V}_t = \mathcal{V}_{t-1} \cup \{w_i^{\star}\}_{i=1}^{M_t}. \tag{7}$$

Note that region partitioning and waypoint selection rely solely on the VLM's strong scene understanding and reasoning via prompting (Appendix A), without any hand-crafted criteria, enabling zero-shot generalization across diverse scenes. Please refer to the Appendix for details on the prompts and the reasoning procedure.

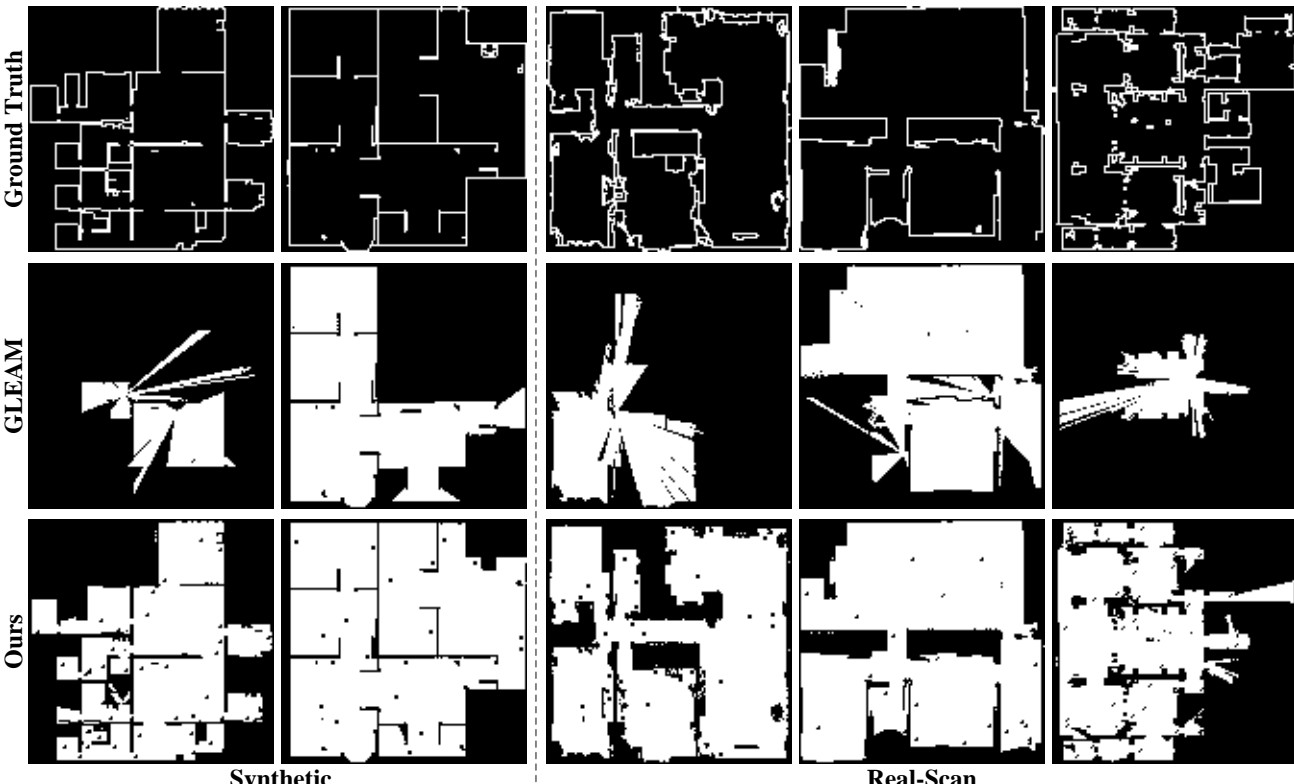

*Figure 6.* The qualitative results of our method and GLEAM on the unseen test set of GLEAM-Bench. Our method consistently outperforms the GLEAM, demonstrating superior performance even in complex scenes.

The exploration procedure is illustrated in Fig. 4. The agent prioritizes executing the nearest VLM-selected waypoint from the current step that advances the current branch toward deeper parts of the scene, following a depth-first policy. All other unvisited VLM-selected waypoints from the current step are cached in a global BEV memory to support efficient backtracking. As exploration proceeds, cached unvisited waypoints that are no longer near frontiers are pruned. When the current branch contains no explorable regions, the agent backtracks to the nearest unvisited node and repeats until the scene is sufficiently explored. Because candidate waypoints at each step are generated from the incremental BEV $\Delta B_t$ and restricted to visible frontiers along the current branch, this design naturally supports a depth-first exploration policy. Local motion between waypoints is planned using a A* planner (Hart et al., 1968).

Although real layouts may contain topological cycles, the frontier-based pruning inherently prevents looping: waypoints in already-explored areas are removed from the candidate set, and revisiting an explored waypoint (the prerequisite for closing a cycle) is therefore disallowed. The policy is also self-correcting: new candidate waypoints are regenerated from the incremental BEV at every step, providing the VLM with fresh observations and opportunities to correct prior errors.

Leveraging the scene topology tree built during active mapping and our VLM-based instruction pipeline, the proposed method facilitates downstream human–agent interaction, such as navigating to a user-specified region and executing language-specified tasks. Users first choose a coarse localization near the target region via tree nodes, then provide a brief description to precisely specify the target region. This yields concise and precise instructions without requiring lengthy or complex prompts to specify the path to the coarse localization (see Fig. 7). Please refer to the Appendix for details on the prompts and the reasoning procedure.

## 4. Experiment

### 4.1. Experimental Setup

**Dataset and Baselines.** We evaluate zero-shot generalization of our method on the recently proposed GLEAM-Bench, which is the first large-scale exploration dataset that includes both synthetic scenes and real-scan scenes. Because GLEAM-Bench is designed to assess mapping coverage and exploration efficiency, it provides textureless scenes to reduce storage and accelerate simulation. Textureless images also help preserve privacy. Accordingly, we use same textureless observations in our experiments. For baseline comparisons, we report the official numbers provided by

*Table 1.* The comparative results on the GLEAM-Bench, including synthetic scenes and real-scan scenes. For baseline, we report the official numbers provided by GLEAM-Bench.

| Method | | Overall | | | Synthetic Scenes | | | Real-Scan Scenes | | |
|---|---|---|---|---|---|---|---|---|---|---|
| | | Cov. ↑ | AUC ↑ | CD ↓ | Cov. ↑ | AUC ↑ | CD ↓ | Cov. ↑ | AUC ↑ | CD ↓ |
| Heuristic | Vacuum (Chen et al., 2019) | 42.96% | 35.89% | 1.67m | 45.81% | 37.56% | 1.41m | 37.13% | 32.48% | 2.20m |
| | FBE (Yamauchi, 1997) | 56.80% | 45.07% | 0.94m | 61.56% | 50.99% | 0.56m | 46.17% | 37.60% | 1.72m |
| Info | UPEN (Georgakis et al., 2022) | 49.65% | 42.98% | 1.38m | 54.84% | 46.65% | 1.01m | 39.02% | 35.47% | 2.14m |
| | ANM (Yan et al., 2023) | 57.01% | 49.56% | 1.02m | 63.98% | 56.37% | 0.66m | 42.73% | 35.63% | 1.77m |
| RL-based | ANS (Chaplot et al., 2020) | 48.86% | 41.98% | 1.39m | 54.31% | 46.93% | 1.01m | 37.71% | 31.86% | 2.17m |
| | OccAnt (Ramakrishnan et al., 2020) | 53.61% | 46.03% | 1.16m | 60.30% | 51.99% | 0.82m | 39.91% | 33.84% | 1.85m |
| | GLEAM (Chen et al., 2025b) | 66.50% | 57.63% | 0.80m | 76.01% | 66.13% | 0.38m | 47.04% | 40.23% | 1.67m |
| VLM-based | **Ours (Zero-shot)** | **75.31%** | **65.70%** | **0.75m** | **83.82%** | **73.54%** | **0.31m** | **57.90%** | **49.64%** | **1.64m** |

*Table 2.* The comparative results on exploration efficiency.

| Methods | FBE | UPEN | ANM | OccAnt | GLEAM | Our |
|---|---|---|---|---|---|---|
| Coverage | 56.80% | 49.65% | 57.01% | 53.61% | 66.50% | **75.31%** |
| Traj. Length | 52.34m | 30.45m | 46.51m | 38.08m | 54.51m | **52.03m** |

*Table 3.* The one-step inference time of our key components.

| Components | VLM inference | Representation | Others |
|---|---|---|---|
| **Time** | 1.38s | 0.06s | 0.02s |

GLEAM-Bench.

**Evaluation Metrics.** We follow the evaluation metrics of GLEAM-Bench, focusing on mapping coverage and exploration efficiency: (1) **Coverage** (Cov., %) measures the fraction of the environment that is successfully mapped at the end of the exploration process. (2) **AUC** (%) computes the area under the coverage-time curve over the entire exploration duration, reflecting both path efficiency and exploration thoroughness. (3) **Chamfer Distance** (CD, meters) is the mono-directional Chamfer distance between each ground-truth point and the nearest captured points.

**Implementation Details.** Following GLEAM-Bench, all experiments are conducted in NVIDIA Isaac Gym (Makoviychuk et al., 2021) with legged gym. We use a CrazyFlie (Giernacki et al., 2017) quadrotor as the agent, equipped with onboard cameras. We render 360° observations at $700 \times 1400$ resolution. The resolution of BEV is $128 \times 128$. In the candidate waypoints generation module, we set $K{=}25$, $d_{\min}{=}2$, $r_{\min}{=}4$, $d_f{=}10$, and $r_p{=}25$. For the agent model, we use the Qwen3-vl-235b-a22b-instruct API (Bai et al., 2025) with temperature$= 0$, seed$= 100$, and max_tokens$= 2000$. The prompt templates used to drive the agent are provided in the Appendix.

### 4.2. Comparison

To comprehensively evaluate zero-shot generalization, we report results averaged over three independent evaluations. As shown in Table 1 and Fig. 6, without any task-specific training or fine-tuning, our method consistently surpasses all baselines, demonstrating strong zero-shot generalization. In particular, our method achieves 83.82% coverage in unseen synthetic indoor environments, averaging more than five rooms. Moreover, on challenging real-scan scenes, it still attains approximately 57.90% final coverage. Notably, our method surpasses GLEAM by 13.25% and 14.00% in coverage and AUC across all scenes. To evaluate the stability of the proposed method, we conduct three independent evaluations and calculate the sample variance, with the results presented in Table 8. The small variance values of the proposed method demonstrate its high stability.

Our method attains the highest AUC on all scenes, indicating more efficient exploration and better information utilization. To further assess exploration efficiency, we quantitatively compare the trajectory lengths. As shown in Table 2, our method yields shorter trajectories than GLEAM while achieving substantially higher coverage, suggesting minimal redundancy and high exploration efficiency. We also compare the inference time of the core components of our method as shown in Table 3, and observe that the VLM incurs substantially higher latency than the other components. Since the current VLM is accessed via the API, local deployment should yield notable speedups.

### 4.3. Human–Agent Interaction

To assess whether the proposed method facilitates human–agent interaction, we design qualitative experiments on the scene map maintenance task executed after active mapping. In particular, when a subset of the scene undergoes layout changes (e.g., a renovated living room or bedroom), the agent is required to update only the affected region of the map rather than re-scan the entire scene. By leveraging our VLM-based instruction pipeline and the scene topology tree, users can compose concise and precise language instructions to specify the target region for updating. As illustrated in Fig. 7, the agent reliably follows these instructions, navigates to the target region, and updates the corresponding map. Beyond scene map maintenance, the proposed method readily transfers to other language-guided navigation task, enabling the agent to reach arbitrary user-specified locations via natural-language instructions. Please refer to the

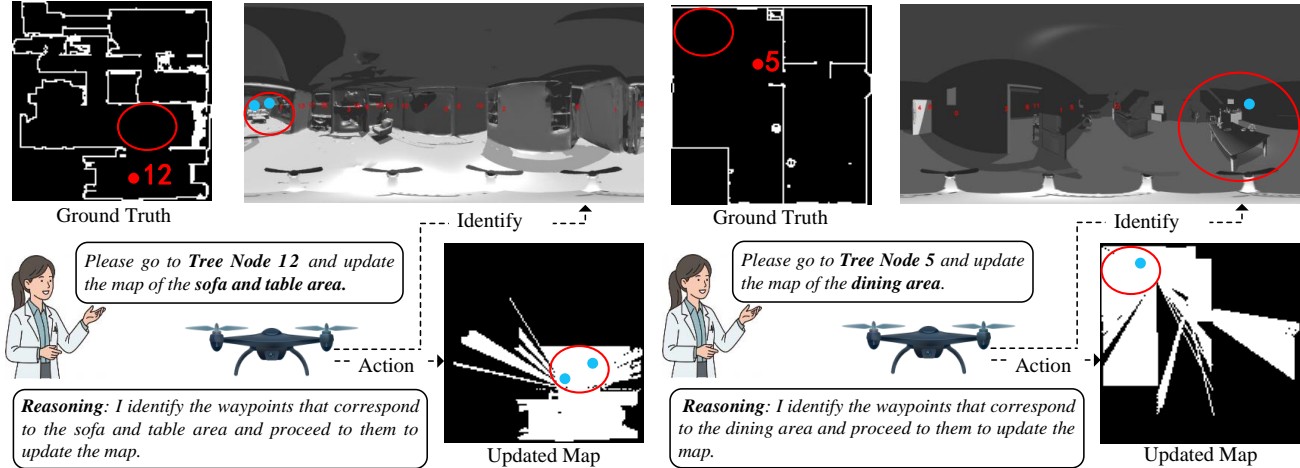

*Figure 7.* Illustration of language-driven human–agent interaction. A user issues natural-language commands grounded to nodes in the scene topology tree to specify target regions, and the agent follows these instructions to update the corresponding areas. Red circles denote language-specified regions, red dots mark nodes in the scene topology tree, and blue dots indicate the waypoints selected by the agent in response to the instruction.

Appendix B.4 for the detailed reasoning procedure.

## 4.4. Ablation Study

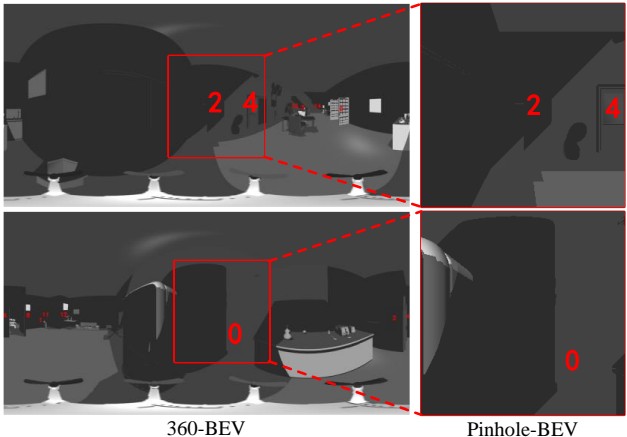

*Figure 8.* Qualitative comparison of 360-BEV and pinhole-BEV. The 360-BEV contains substantially more scene information than the pinhole-BEV.

**The Effect of Fused 360-BEV Representations.** 360-BEV enhances omnidirectional perception in the VLM, yielding superior active mapping performance. To evaluate its effectiveness, we conduct an ablation study in which we swap 360-BEV for Pinhole-BEV in scene representation, while keeping the rest of the pipeline unchanged. As shown in Table 4, 360-BEV substantially improves mapping coverage relative to Pinhole-BEV (75.31% vs. 38.77%), highlighting the critical importance of omnidirectional perception for active mapping. The qualitative comparisons (Fig. 8) show that 360-BEV contains substantially more

*Table 4.* Ablation study on scene representation and candidate waypoints generation.

| Settings | Cov. ↑ | AUC ↑ | CD ↓ |
|---|---|---|---|
| **Scene Representation** | | | |
| Pinhole-BEV | 38.77% | 28.46% | 2.21m |
| **360-BEV (Our)** | **75.31%** | **65.70%** | **0.75m** |
| **Candidate Waypoints** | | | |
| Coordinates | 45.10% | 40.75% | **1.93m** |
| **IDs (Our)** | **75.31%** | **65.70%** | **0.75m** |

scene information, whereas pinhole-BEV overlooks many high-value regions during exploration.

**The Effect of Candidate Waypoints Generation.** We associate semantics from 360 image with BEV geometry by generating a set of image-space candidate waypoints, each tagged with a numeric identifier that uniquely binds to the BEV coordinate. This design enables the VLM to operate entirely in its preferred image space, while outputting precise metric actions in 3D space via 360-to-BEV mapping. To validate effectiveness, we perform an ablation in which the VLM directly predicts BEV-space coordinates to produce 3D actions, while keeping the rest of the pipeline unchanged. As shown in Table 4, our method achieves substantially higher coverage (75.31% vs 45.10%), demonstrating that decision-making in the image space better leverages VLM capabilities. Moreover, the coordinates directly predicted by the VLM are concentrated at the regional centers, neglecting high-value marginal areas and thus leading to under-exploration. In contrast, our strategy generates spatially uniform waypoints, enabling the agent to achieve more thorough exploration.

*Table 5.* Ablation study on VLM-Based zero-shot exploration.

| Settings | Cov. ↑ | AUC ↑ | CD ↓ |
|---|---|---|---|
| Qwen3-vl-8b-instruct | 50.35% | 47.39% | 1.53m |
| **Qwen3-vl-235b-a22b-instruct (Our)** | **75.31%** | **65.70%** | **0.75m** |

**The Effect of VLM-Based Zero-Shot Exploration.**
Our VLM-based exploration agent leverages large vision–language models to enable zero-shot active mapping. To quantify how model capacity affects zero-shot performance, we conduct an ablation study that swaps the VLM while keeping the rest of the pipeline unchanged. As shown in Table 5, replacing Qwen3-vl-8b-instruct with the higher-capacity Qwen3-vl-235b-a22b-instruct yields a substantial improvement in coverage (75.31% vs. 50.35%). We attribute these gains to the larger model's stronger semantic grounding and reasoning, which translate into more informative waypoint selection and fewer redundant scans. These results indicate that our approach scales with VLM capability, with larger models delivering greater generalization and exploration efficiency.

**The Effect of VLM-Based Waypoint Selection.** A natural question is whether the VLM contributes beyond the geometric selection of candidate waypoints. To isolate its effect, we keep the entire pipeline fixed and replace the VLM selector with two geometry-only heuristics: *Nearest-to-Centroid*, which picks the waypoint closest to the current region centroid, and *Max-Frontier-Distance*. As shown in Table 6, the VLM-based selector substantially outperforms both heuristics. The gain stems from the VLM's semantic understanding: it identifies nearly all transition points (e.g., doors and corridors) that open onto large unexplored areas, whereas geometry-only heuristics lack such scene-level semantics and frequently miss these passages, leading to under-exploration.

**The Effect of Depth-First Exploration.** To validate the contribution of the depth-first exploration policy, we vary only the exploration policy while all other components are kept unchanged. As shown in Table 7, our method attains the best coverage-efficiency balance. Breadth-first exploration reaches coverage comparable to ours (73.82% vs. 75.31%) but requires far longer trajectories (82.77m vs. 52.03m), because it must visit all sibling nodes at each depth before descending, forcing repeated traversal between distant regions. In contrast, our method explores each branch deeply before backtracking, achieving high coverage with efficient trajectories. This advantage is further amplified by the scene topology tree memory: every VLM-selected waypoint is cached as a node and pruned once it leaves the frontier, so backtracking reduces to selecting the nearest unexplored node rather than replanning, keeping the cost of leaving a dead-end branch low.

*Table 6.* Ablation on the waypoint selection strategy.

| Waypoint Selection | Cov. ↑ | AUC ↑ |
|---|---|---|
| Nearest-to-Centroid | 47.96% | 45.93% |
| Max-Frontier-Distance | 52.85% | 50.92% |
| VLM-based (Ours) | **75.31%** | **65.70%** |

*Table 7.* Ablation on the exploration policy. All other components are kept unchanged.

| Policy | Cov. ↑ | Traj. Length ↓ |
|---|---|---|
| Breadth-first exploration | 73.82% | 82.77m |
| Ours | **75.31%** | **52.03m** |

## 5. Conclusion

We introduced a VLM-based active mapping method that achieves zero-shot mapping in large-scale scenes and facilitates language-driven human–agent interaction. Our method proposes a fused 360–BEV representation for omnidirectional semantic–geometric perception. To enable VLM decision-making in its preferred image space while outputting actions in embodied 3D space, we design a candidate waypoint generation strategy. Finally, we design a VLM-based exploration agent that organizes scenes into the scene topology tree and employs a depth-first exploration policy, achieving thorough coverage in large-scale scenes. On large-scale benchmarks, our zero-shot method significantly surpasses all baselines. Moreover, by leveraging the VLM-based instruction pipeline and the scene topology tree, users can compose concise and precise language instructions that enable effective human–agent interaction. Our contribution is not the individual primitives but the design principles that unlock a frozen VLM for embodied active mapping: which representation best fuses semantics and geometry for a VLM, how to convert image-space VLM decisions into executable 3D actions, and how to structure exploration with tree-based memory.

## 6. Limitations

We identify two principal limitations. The limited BEV resolution can cause frontier detection to overlook small openings, so a connected region may stay undiscovered until observed from another vantage point. The VLM occasionally misses a waypoint leading to unexplored space, or over-segments a contiguous area into spurious regions, producing redundant steps.

## Impact Statement

This paper presents work whose goal is to advance the field of Machine Learning. There are many potential societal consequences of our work, none which we feel must be specifically highlighted here.

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

## A. Prompt Templates

We design prompt templates for zero-shot active mapping and language-driven human-agent interaction (Figs. 9 and 10). Given the requirement for the agent to interact with human users, we adopt a two-step prompting scheme. (1) The agent parses user instructions and extracts the scene tree nodes corresponding to coarse locations and the description delineating the goal region. (2) The agent arrives at the scene tree nodes, identifies key waypoints for the goal region based on the extracted description, and navigates to the target location to complete the task.

---

**VLM-Based Zero-Shot Active Mapping**

*You are an active mapping agent. The input is an image, in which multiple candidate waypoints are annotated with unique IDs. These candidate waypoints correspond to potential viewpoints for exploration and reconstruction, and are distributed across different regions of the scene.*

*Waypoints are selected according to the following procedure: (1) Analyze the spatial layout of different regions in the image and establish correspondences between regions in the image and the candidate waypoints to support subsequent decision-making. (2) Assign higher exploration value to candidate waypoints that provide access to a larger number of unexplored areas. (3) For each region, select only one candidate waypoint with the highest exploration value.*

*You should carefully select waypoints that follow the defined waypoint procedures, and return a JSON object with the field "Waypoints", whose value is a list containing only the IDs of the selected waypoints, ranked in descending order of their IDs. You are required to conduct step-by-step reasoning without skipping any steps, and perform a self-check on the reasoning and output.*

---

*Figure 9.* Prompt templates for VLM-based zero-shot active mapping.

---

**Language-Driven Human–Agent Interaction**

**Users**: *please go to {scene tree nodes} and execute the {specific tasks} within the {descriptions of target regions}.*

**Agent:** *follow the user's instructions in two steps.*

1. Instruction Understanding:

*Extract the IDs of {scene tree nodes} and the {descriptions of target regions} based on the instruction.*

2. Upon arrival at the coarse localization corresponding to the **{scene tree nodes}**, identify the waypoints that correspond to the *{descriptions of target regions}*:

*The input is an image, in which multiple candidate waypoints are annotated with unique IDs. These candidate waypoints correspond to potential viewpoints for exploration, and are distributed across different regions of the scene. Waypoints are selected according to the following procedure: (1) Analyze the spatial layout of rooms in the image and establish correspondences between rooms and candidate waypoints to support subsequent decision-making. (2) Select the key candidate waypoints that correspond to the {descriptions of target regions}.*

*You should carefully select waypoints that follow the defined waypoint procedures, and return a JSON object with the field "Waypoints", whose value is a list containing only the IDs of the selected waypoints, ranked in descending order of their IDs. You are required to conduct step-by-step reasoning without skipping any steps, and perform a self-check on the reasoning and output.*

---

*Figure 10.* Prompt templates for language-driven human–agent interaction.

## B. More Results

### B.1. Stability Evaluation

To evaluate the stability of the proposed method, we conduct three independent evaluations and calculate the sample variance, with the results presented in Table 8. The small variance values of the proposed method demonstrate its high stability.

*Table 8.* The sample variance across three evaluations.

| Metrics | Cov. ↑ | AUC ↑ | CD ↓ |
|---------|--------|-------|------|
| **Variance** | 0.4693 | 0.1252 | 0.0030 |

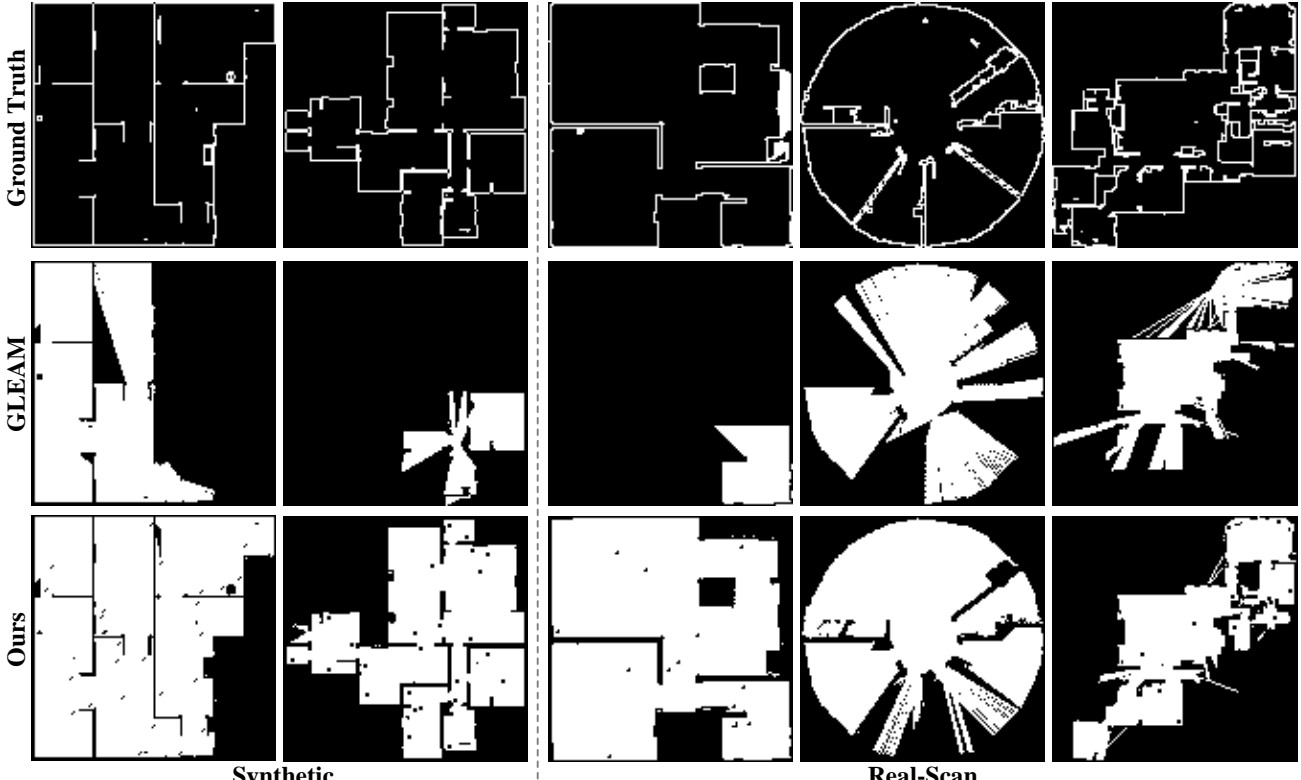

**Synthetic**  **Real–Scan**

*Figure 11.* The qualitative results of GLEAM and our method on the unseen test set of GLEAMBench. Our method consistently outperforms the GLEAM, demonstrating superior performance.

### B.2. Generalization across VLM Backbones.

Keeping the pipeline fixed, we swap the agent VLM. As shown in Table 9, our framework remains competitive across diverse VLM families and still surpasses the trained baseline GLEAM (66.50% Cov.), indicating the gains come from the framework design rather than from a single ultra-large backbone.

### B.3. More Qualitative Comparisons

We provide more qualitative comparisons in Fig. 11, showing that our zero-shot method substantially outperforms the strongest baseline that requires large-scale training data.

### B.4. Detailed Reasoning

To supplement the agent's reasoning details presented in Figs. 5 and 7 of the main paper, we present the detailed reasoning for region partitioning and informative waypoint selection in Figs. 12 and 13, and the detailed reasoning for language-driven human–agent interaction in Figs. 14 and 15.

*Table 9.* Generalization across VLM backbones within our framework.

| VLM Backbone | Cov. ↑ | AUC ↑ | CD ↓ |
|---|---|---|---|
| Gemini-2.5-flash | 71.73% | 63.07% | 0.95m |
| GPT-4o | 72.47% | 63.85% | 0.83m |
| Qwen3-VL-235B-A22B (Ours) | **75.31%** | **65.70%** | **0.75m** |

**Region Partitioning and Informative Waypoint Selection**

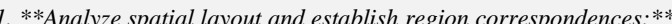

```
{
  "Waypoints": [24, 9, 7]
}
```

**Step-by-step Reasoning:**

1. **Analyze spatial layout and establish region correspondences:**
   - The panoramic image shows a large interior space with multiple distinct regions that appear to be separate regions.
   - On the left side of the image, there is a lounge-like area with multiple sofas and tables — this region contains waypoints 9, 5, and 6. These are clustered together, suggesting they belong to the same region.
   - In the center, there's a more open area with structural pillars and what appears to be a corridor or transition region — waypoints 24, 23, and 22 are located here. They are spaced apart but within a contiguous central region.
   - On the right side, there is another furnished area with chairs, equipment, and storage units — waypoints 4 and 7 are in this region, likely representing a separate region or workspace.

2. **Assign exploration value based on access to unexplored areas:**
   - Waypoint 24 (center) is positioned near a major pillar and faces toward both the left lounge area and the right workspace. It offers a broad field of view and potential access paths into multiple subregions — thus, it has high exploration value.
   - Waypoint 9 (left) is situated at the edge of the lounge area and provides a clear view into the central corridor and possibly toward the back of the region. It allows exploration deeper into the left wing.
   - Waypoint 7 (right) is located in the workspace area and appears to face toward an entrance or hallway leading to other parts of the building. It offers access to potentially unexplored zones beyond the current frame.
   - Waypoints 5, 6, 23, 22, and 4 are either redundant (too close to higher-value waypoints) or offer limited new visibility compared to their neighbors.

3. **Select one waypoint per region with highest exploration value:**
   - Left region (lounge): Among 9, 5, 6 → **9** has the best vantage point toward the center and rear.
   - Central region: Among 24, 23, 22 → **24** is most centrally located and offers maximal coverage.
   - Right region (workspace): Among 4, 7 → **7** is positioned to look outward, suggesting better access to unexplored areas.

4. **Rank selected waypoints in descending order of IDs:**
   - Selected IDs: 24, 9, 7
   - Sorted descending: [24, 9, 7]

**Self-check:**
- All selected waypoints are from different regions — satisfies condition (3).
- Each selected waypoint offers superior access to unexplored areas compared to others in its region — satisfies condition (2).
- Spatial layout analysis was performed to assign regions correctly — satisfies condition (1).
- Output is in correct JSON format with descending ID order.
- No steps were skipped; reasoning is complete and logically consistent.

*Final output is valid and meets all requirements.*

*Figure 12.* Detailed reasoning for region partitioning and informative waypoint selection in VLM-based agents. Red circles indicate the partitioned regions, and blue circles mark point locations. The IDs of selected points are enlarged for better visibility.

**Region partitioning and informative waypoint selection**

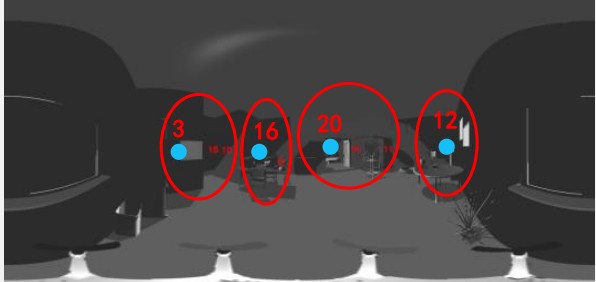

```
{
  "Waypoints": [20, 16, 12, 3]
}
```

**Step-by-Step Reasoning:**

1. **Analyze Spatial Layout and Region Correspondences:**
   - The image appears to be a panoramic or 360-degree view of an interior space with multiple regions.
   - Waypoint 3 is located near a doorway or entrance on the left side — likely a transition point between regions.
   - Waypoint 15 and 10 are in the same general area as 3, but closer to the center; they seem to be inside a main region with furniture (table and chairs).
   - Waypoint 16 is positioned at a central table with chairs — this is likely the main activity area of the central region.
   - Waypoint 0 is very close to 16, possibly redundant for exploration purposes.
   - Waypoint 20 is near what appears to be a doorway or passage leading to another region (possibly a kitchen or utility area), indicated by the presence of a refrigerator-like object and shelving (waypoint 14, 11).
   - Waypoint 14 and 11 are adjacent to 20, suggesting they are within the same secondary region.
   - Waypoint 12 is near a round table with objects on it, possibly in a separate seating or dining zone on the right side of the scene.
   - Waypoint 12 is also near windows or light sources, indicating it may offer views into or out of the region.

2. **Prioritize Transition Points and High Exploration Value:**
   - According to rule (2), transition points (doors, corridors) should be prioritized because they provide access to unexplored areas.
     - Waypoint 20 is clearly near a doorway or opening into another region — highest priority for exploration.
     - Waypoint 3 is also near a doorway or entrance — second-highest priority.
   - For each region, we select only one waypoint with the highest exploration value (rule 3).
     - Central region (with table/chairs): Waypoint 16 offers the best vantage point to survey the region and potentially see transitions to other areas. Waypoint 0 is too close and redundant.
     - Right-side region (with round table): Waypoint 12 provides a clear view of the table and nearby windows/doorways — best representative of that region.
     - Secondary region (near 20, 14, 11): Waypoint 20 is the transition point — already selected for its high exploration value.
     - Left-side region (3, 15, 10): Waypoint 3 is the transition point — already selected.

3. **Final Selection and Ranking:**
   - We select one waypoint per distinct region:
     - Transition to secondary region → 20
     - Central region → 16
     - Right-side region → 12
     - Left-side transition → 3
   - No other waypoints add unique coverage or higher exploration value than these four.
   - Sort by ID in descending order: 20 > 16 > 12 > 3

4. **Self-check:**
   - All selected waypoints correspond to distinct spatial regions.
   - Only one waypoint per region is selected.
   - IDs are sorted in descending order.
   - No redundant or low-value waypoints are included.

All conditions satisfied. Output is valid.

*Figure 13.* Detailed reasoning for region partitioning and informative waypoint selection in VLM-based agents. Red circles indicate the partitioned regions, and blue circles mark point locations. The IDs of selected points are enlarged for better visibility.

**Language-Driven Human–Agent Interaction**

*The input is an image, in which multiple candidate waypoints are annotated with unique IDs. These candidate waypoints correspond to potential viewpoints for exploration, and are distributed across different regions of the scene. Waypoints are selected according to the following procedure: (1) Analyze the spatial layout of regions in the image and establish correspondences between regions and candidate waypoints to support subsequent decision-making. (2) Select the key candidate waypoints that correspond to the {dining area}.*

*{*
*  "Waypoints": [7]*
*}*

*---*

***Step-by-step Reasoning:***

*1. **Analyze the spatial layout of regions and establish correspondences between regions and candidate waypoints:***
*   - The panoramic image shows a 360-degree view of an indoor environment with multiple furniture items and architectural features.*
*   - Waypoint 4 and 8 are near a door or entryway on the far left.*
*   - Waypoint 2 is positioned at what appears to be a doorway leading into another region.*
*   - Waypoints 6 and 11 are located in a transitional space, possibly a hallway or entrance to a living area.*
*   - Waypoint 1 and 5 are adjacent to a TV unit and chair — suggesting a living region or media area.*
*   - Waypoint 12 is next to a sofa — likely part of a living region.*
*   - Waypoint 7 is clearly associated with a large dining table surrounded by chairs and objects like plates and bottles — this strongly indicates a dining area.*

*2. **Select the key candidate waypoints that correspond to the dining area:***
*   - Among all annotated waypoints, only waypoint 7 is unambiguously situated within the dining area, as evidenced by the presence of a dining table and related dining accessories.*
*   - No other waypoint (e.g., 1, 5, 12) is directly associated with dining furniture or context.*

*3. **Rank selected waypoints in descending order of their IDs:***
*   - Since only one waypoint (7) satisfies the condition, the list contains only [7], which is already in descending order.*

*4. **Self-check:***
*   - Does the selected waypoint correspond to the dining area? Yes, waypoint 7 is centered on the dining table.*
*   - Are waypoints ranked in descending order? Only one element, so trivially satisfied.*
*   - Have any irrelevant waypoints been included? No.*

*Final output is correct and meets all specified requirements.*

*Figure 14.* Detailed reasoning for language-driven human–agent interaction in VLM-based agents. Red circles denote language-specified regions, and blue dots indicate the waypoints selected by the agent in response to the instruction.

**Language-Driven Human–Agent Interaction**

*The input is an image, in which multiple candidate waypoints are annotated with unique IDs. These candidate waypoints correspond to potential viewpoints for exploration, and are distributed across different regions of the scene. Waypoints are selected according to the following procedure: (1) Analyze the spatial layout of regions in the image and establish correspondences between regions and candidate waypoints to support subsequent decision-making. (2) Select the key candidate waypoints that correspond to the {sofa and coffee table}.*

*{*
  *"Waypoints": [11, 5]*
*}*

*---*

*\*\*Step-by-step Reasoning:\*\**

*1. \*\*Analyze spatial layout and establish region-waypoint correspondence:\*\**
  *- The image is a panoramic grayscale reconstruction with multiple annotated waypoints (IDs 0–23).*
  *- On the far left side of the panorama, we can clearly see a living region area containing a sofa and a coffee table.*
  *- Waypoint 5 is positioned directly over the sofa.*
  *- Waypoint 11 is located near the coffee table, slightly in front of the sofa.*
  *- These two waypoints are spatially associated with the key furniture items: sofa and coffee table.*

*2. \*\*Select waypoints corresponding to sofa and coffee table:\*\**
  *- Based on visual inspection, waypoint 5 corresponds to the sofa.*
  *- Waypoint 11 corresponds to the coffee table.*
  *- No other waypoints are as clearly or directly associated with these specific objects.*

*3. \*\*Rank selected waypoints in descending order of IDs:\*\**
  *- Selected IDs: 5 and 11.*
  *- Descending order: [11, 5].*

*4. \*\*Self-check:\*\**
  *- Are the selected waypoints associated with the sofa and coffee table? → Yes.*
  *- Are the IDs ranked in descending order? → Yes.*
  *- Are all IDs valid (present in the image)? → Yes.*

*All conditions satisfied. Output is correct.*

*Figure 15.* Detailed reasoning for language-driven human–agent interaction in VLM-based agents. Red circles denote language-specified regions, and blue dots indicate the waypoints selected by the agent in response to the instruction.

