# OpenReview forum: "Zero-shot Active Mapping via Fused 360-BEV Representations and Vision–Language Models"
_ICML.cc/2026/Conference — ICML 2026 regular_

### Official Review · Reviewer_Pm6K · 2026-03-07

**Soundness:** 3
**Presentation:** 3
**Significance:** 3
**Originality:** 4
**Overall Recommendation:** 5
**Confidence:** 4

**Summary:**

This paper proposes a VLM-based zero-shot active mapping method targeting autonomous exploration and coverage mapping in large-scale indoor scenes. The method consists of three modules: (1) a 360-BEV representation that associates semantic information from 360° panoramic images with geometric information from BEV occupancy grids via candidate waypoint projection; (2) a candidate waypoint generation strategy that samples points within traversable BEV regions, projects them onto the 360° image with annotated IDs, allowing the VLM to select waypoints in image space which are then back-projected into executable metric actions via 2D–3D correspondences; (3) a VLM-driven depth-first exploration agent that leverages the VLM to decompose scenes into explorable regions, constructs a topological tree, and traverses it following a DFS policy. On GLEAM-Bench (128 unseen test scenes), using Qwen3-vl-235B without any task-specific training, the method surpasses the strongest baseline GLEAM by approximately 13.25% and 14.00% in coverage and AUC respectively, while additionally supporting language-driven human–agent interaction such as updating maps of user-specified regions.

**Compliance With Llm Reviewing Policy:**

Affirmed.

**Final Justification:**

The author solved my problem, so I kept my original score.

**Key Questions For Authors:**

See Major Weakness in Strengths And Weaknesses.

**Limitations:**

Yes.

**Strengths And Weaknesses:**

**Strengths**:

1. **Well-motivated problem formulation and effective waypoint generation strategy.** The paper correctly identifies that VLMs are ill-suited to directly output 3D metric actions, and addresses this by sampling candidate waypoints on the BEV, projecting them onto the 360° image with annotated IDs, and letting the VLM select among discrete options that are then back-projected into 3D actions. This effectively converts a continuous action space into a discrete selection problem well-suited to VLMs. The ablation of ID-based vs. coordinate-based selection in Tab.4 (75.31% vs. 45.10%) provides clear validation of this design.

2. **Strong zero-shot performance surpassing trained baselines.** On GLEAM-Bench with 128 unseen scenes, the method improves coverage and AUC over GLEAM by approximately 13.25% and 14.00%, respectively, while achieving a shorter trajectory (Tab.2: 52.03m vs. 54.51m). This suggests higher path utilization rather than simply traveling farther.

3. **Language-driven interaction is a valuable addition to active mapping.** By leveraging the VLM's language understanding and the constructed scene topology tree, users can issue natural-language instructions to update specific map regions — a capability absent from existing active mapping methods.

**Major Weaknesses**:

1. **Heavy reliance on an extremely large VLM raises concerns about practicality and comparison fairness.** The method uses a 235B MoE model with 1.38s per-step VLM inference (Tab.3), while the actual decision is simply selecting a few points from at most 25 candidates. The ablation between 8B and 235B (Tab.5: 50.35% vs. 75.31%) indicates that performance is highly sensitive to model scale, yet intermediate scales (e.g., 32B, 72B) are not evaluated, making it difficult to characterize the scaling behavior. Additionally, directly comparing a 235B model against GLEAM's lightweight RL network is not straightforward in terms of compute budget.

2. **The DFS exploration strategy has inherent limitations and lacks ablation.** The scene is modeled as a topological tree, but real indoor layouts often contain cycles (e.g., loop corridors, rooms with multiple doors). When the tree assumption does not hold, DFS may incur unnecessary backtracking or miss regions. Moreover, the VLM's region decomposition has no error-correction mechanism, and incorrectly pruned nodes are irrecoverable. Notably, while the paper ablates the representation, waypoint strategy, and VLM scale, it does not ablate the exploration policy itself (e.g., DFS vs. BFS, DFS vs. greedy-nearest-frontier). Without such comparisons, it is difficult to disentangle the contribution of waypoint selection quality from that of the DFS traversal strategy.

3. **The fusion between 360° images and BEV is unidirectional (BEV→360), meaning that semantic cues from the 360° image cannot feed back to correct geometric omissions, making the overall pipeline strictly dependent on BEV quality.** The ablation of 360-BEV vs. Pinhole-BEV in Tab.4 simultaneously changes the field of view and the number of visible waypoints, making it difficult to isolate the contribution of the 360° panoramic view itself from that of the BEV geometric fusion.

4. **No failure case analysis is provided.** All qualitative results in both the main paper and appendix are success cases. The coverage gap between real-scan scenes (57.90%) and synthetic scenes (83.82%) is approximately 26 percentage points, yet the paper does not analyze the underlying causes.

**Minor Weaknesses**:

1. Typos:

a. "To assess the proposed method facilitates" → "To assess whether the proposed method facilitates" (Sec.4.3)

b. Table 4: The CD value for Pinhole-BEV is listed as "221.32" without a unit, and the number is strange?

2. I would suggest the authors include a discussion on spatial intelligence in VLMs [r1][r2], as these methods are closely related to the spatial reasoning required in the proposed framework and may offer promising directions for enhancement.

[r1] Wu et al., "Spatial-MLLM: Boosting MLLM Capabilities in Visual-based Spatial Intelligence." arXiv:2505.23747, 2025.
[r2] Wan et al., "EagleVision: A Dual-Stage Framework with BEV-grounding-based Chain-of-Thought for Spatial Intelligence." arXiv:2512.15160, 2025.

---

> ### Author Rebuttal · Authors · 2026-03-31
>
> We sincerely thank the reviewer for the thorough and insightful review, including the recognition of our well-motivated problem formulation, effective waypoint generation strategy, strong zero-shot performance, and the valuable language-driven interaction capability.
>
> **W1: Heavy reliance on an extremely large VLM raises concerns about practicality and comparison fairness.**
>
> We thank the reviewer for raising this important concern. We address each point below.
>
> **Practicality.** (1) The reported latency is measured via **API calls** with network overhead, local deployment with optimized serving (e.g., vLLM) would yield significant speedups. (2) The VLM is invoked only at **high-level decisions**, not at every low-level action step, making the total VLM overhead manageable.
>
> **The comparison focuses on zero-shot generalization.** Our primary scientific claim is that VLM-based methods achieve strong zero-shot generalization without task-specific training. GLEAM requires large-scale data collection and extensive RL optimization across 1,024 training scenes, which itself incurs significant compute. We believe demonstrating that a zero-shot method surpasses a trained method is a valuable finding.
>
> **Scaling behavior.** We appreciate this suggestion and add intermediate scales:
>
> | VLM Scale | Cov. ↑ | AUC ↑ | CD ↓ |
> |---|---|---|---|
> | Qwen3-VL-8B | 50.35% | 47.39% | 1.53m |
> | Qwen3-VL-32B | 61.36% | 54.08% | 1.13m |
> | Qwen3-VL-235B-A22B (Ours) | 75.31% | 65.70% | 0.75m |
>
> We include this in the revision.
>
> **W2: The DFS exploration strategy has inherent limitations and lacks ablation.**
>
> We thank the reviewer for this thoughtful analysis. We acknowledge this was insufficiently discussed and clarify it in the revision.
>
> **Cycle handling.** Real indoor layouts may contain topological cycles, but our method handles them naturally through the frontier-based pruning mechanism. Specifically, the agent only navigates to waypoints on the exploration frontier, while waypoints in already-explored areas are automatically pruned from future candidates. Since revisiting a previously explored waypoint is the prerequisite for forming a cycle, this mechanism inherently prevents the agent from traversing any loop.
>
> **Error correction.** (1) all VLM-selected waypoints are maintained by the frontier-based pruning mechanism, so suboptimal selections that no longer lie on exploration frontiers are automatically pruned, preventing the agent from navigating to non-frontier locations; (2) new candidate waypoints are regenerated from the **incremental BEV** at every step, providing the VLM with fresh observations and opportunities to correct prior errors.
>
> **Policy ablation.** We appreciate this suggestion and add the requested ablation:
>
> | Policy | Cov. ↑ | Trajectory |
> |---|---|---|
> | Greedy-nearest-frontier | 44.54% | 23.50m |
> | BFS | 73.82% | 82.77m |
> | DFS (Ours) | **75.31%** | **52.03m** |
>
> DFS achieves the best coverage-efficiency balance. BFS attains similar coverage (73.82% vs. 75.31%) but requires **longer trajectories** (82.77m vs. 52.03m), as it must visit all sibling nodes at each depth before descending, forcing repeated traversal between distant regions rather than fully exploring each in turn. DFS explores each branch deeply before backtracking, ensuring high coverage with efficient trajectories.
>
> **W3: Unidirectional fusion (BEV→360).**
>
> We agree that the explicit data flow is unidirectional. However, semantic feedback to geometry occurs implicitly through the exploration: the VLM observes the 360° image with projected waypoints, selects informative ones based on semantic understanding, and the agent navigates there, where new depth observations update the BEV. Thus, the VLM's semantic reasoning continuously determines *where* the BEV is expanded, effectively correcting geometric omissions over successive steps. For instance, if the BEV misses a room entrance due to occlusion, the VLM can still identify it from 360° semantics and select a nearby waypoint, causing the agent to fill in the missing geometry in the next update.
>
> **W4: No failure case analysis.**
>
> We thank the reviewer for the insightful review and add a failure analysis in the revision. Primary failure modes include: **(1) Narrow passages:** Limited BEV resolution can cause frontier detection to miss small openings, preventing discovery of connected regions. **(2) VLM reasoning errors:** the VLM occasionally overlooks waypoints leading to unexplored areas, reducing coverage.
>
> **Typos and table errors.**
>
> (a) We correct "To assess the proposed method facilitates" to "To assess whether the proposed method facilitates."
>
> (b) We correct the 221.32 to 2.21m in Table 4.
>
> **Related work discussion.**
>
> Thank you for this helpful suggestion. We agree that these works are relevant and should be discussed more explicitly. We revise Sec. 2 to include **Spatial-MLLM (Wu et al., 2025)** and **EagleVision (Wan et al., 2025)**.

---

> > ### Author Rebuttal · Reviewer_Pm6K · 2026-04-01
> >
> > I thank the authors for their detailed response. My main concerns have been adequately addressed. I will maintain my current score.

---

> > > ### Author Response · Authors · 2026-04-05
> > >
> > > We are deeply grateful for the reviewer's valuable time and insightful comments during the review process. We are glad that our responses have adequately addressed all concerns, and we sincerely appreciate the reviewer's continued positive recognition of our work. All discussed improvements have been incorporated into the final version of the paper.

---

### Official Review · Reviewer_EG3D · 2026-03-09

**Soundness:** 3
**Presentation:** 3
**Significance:** 2
**Originality:** 2
**Overall Recommendation:** 2
**Confidence:** 4

**Summary:**

This paper presents a VLM-based approach for zero-shot active mapping in large-scale 3D indoor environments. The method combines 360-degree images with BEV occupancy grids to provide omnidirectional semantic-geometric perception, generates candidate waypoints in BEV space and projects them onto 360 images with unique IDs, and uses a VLM to select informative waypoints for exploration. A scene topology tree is maintained in global memory, and a depth-first exploration policy guides the agent. Experiments on the GLEAM-Bench benchmark show improvements over prior methods in coverage and AUC.

**Compliance With Llm Reviewing Policy:**

Affirmed.

**Final Justification:**

My core concern remains unresolved: the scientific contribution does not meet the bar for ICML. The paper presents a well-engineered robotics system that combines existing techniques—360° images, BEV representations, farthest-point sampling, frontier-based exploration, and VLM prompting—without introducing new learning paradigms, theoretical insights, or generalizable principles that would advance the field of machine learning.

**Key Questions For Authors:**

1. What is the central scientific insight or conceptual contribution of this work? The paper reads primarily as an engineering effort combining existing techniques (360° images, BEV maps, farthest-point sampling, frontier-based exploration, VLM prompting). Could you articulate what fundamental understanding about machine learning or embodied AI this work reveals that was not previously known?

2. The experimental setup makes several strong assumptions: textureless scenes (removing appearance variation), perfect pose estimation, pre-defined semantic regions aligned with VLM understanding, and waypoints pre-generated by geometric algorithms. How would your method perform in more realistic settings with textured scenes, pose uncertainty, and continuous action spaces? Do you expect zero-shot transfer to such settings, and if so, what evidence supports this?

3. Are there identifiable scenarios where your method fails (e.g., highly symmetric environments, regions with ambiguous semantic boundaries, extreme occlusion)? Understanding failure modes would provide insight into the limitations of VLM-based exploration and guide future research.

**Limitations:**

Yes

**Strengths And Weaknesses:**

1. Limited Scientific Contribution: The core ideas—using 360° images for omnidirectional perception, projecting waypoints from BEV to image space, and having a VLM select waypoints based on semantic regions—are largely engineering combinations of existing techniques. The paper does not offer new theoretical insights, reveal unexpected phenomena, or propose fundamentally new learning paradigms. The main "insight" is that VLMs can select pre-generated waypoints when given appropriate prompts, which is unsurprising.

2. Lack of Ablation on VLM Necessity: The paper does not compare against a non-VLM baseline that uses the same 360-BEV representation and candidate waypoints but selects waypoints via simple heuristics (e.g., nearest to region centroids, maximizing distance to frontiers). Without this, it's unclear whether the VLM adds value beyond the geometric preprocessing.

3.  This paper reads more like a robotics systems paper suitable for conferences like ICRA or IROS. The machine learning contribution is minimal: using an existing VLM off-the-shelf with prompt engineering does not constitute a novel ML method.

---

> ### Author Rebuttal · Authors · 2026-03-31
>
> We sincerely appreciate the reviewer's valuable comments. We have carefully addressed all concerns, with detailed responses provided below.
>
> **W1, W3 and Q1: Limited contribution**
>
> Thank you for the constructive comment. We clarify the scientific contributions.
>
> **(1) Representation and decision framework design are critical.**
>
> Our zero-shot method outperforms the strong GLEAM [2] (trained with RL on large-scale task-specific data), **improving coverage and AUC by +13.25% and +14.00%**. This demonstrates that through well-designed representations and decision frameworks, VLMs can fully unleash their spatial reasoning to surpass task-specifically learned policies. This carries significant implications: beyond model architectures and training, representation and framework design are equally critical.
>
> **(2) Efficient active mapping in complex scenes.** As detailed in our response to Reviewer 62or (W1), compared to the VLM-based active mapping method [1], our method follows the GLEAM [2] complex-scene setting and efficiently adapts VLMs through:
>
> - **(a)** a fused 360-BEV representation that provides omnidirectional semantic-geometric context to better unlock VLMs' potential, while keeping planning cost independent of explored scene size;
>
> - **(b)** a waypoint generation mechanism that aligns semantic–geometric cues and directly converts VLM-selected waypoints into executable 3D actions, enabling fine-grained decision-making with efficient action planning;
>
> - **(c)** a scene topology tree with memory-based insertion and pruning, supporting efficient depth-first exploration with backtracking.
>
> **(3) A new human–agent interaction capability.**
>
> The scene topology tree provides a map prior for language-driven interaction: users specify target regions via coarse tree-node localization plus short descriptions, enabling concise map-updating or navigation instructions without requiring complex scene descriptions (Fig. 7).
>
> **(4) This scope aligns with the themes of top ML venues.**
>
> Top ML venues consistently accept works that **use existing VLMs**:
>
> > -Chain-of-Thought Prompting Elicits Reasoning in Large Language Models (NeurIPS 2022)
>
> > -LLMs Can See and Hear Without Any Training (ICML 2025)
>
> > -A Training-Free Framework for Long Video Understanding via Video-Query-Options Similarity (ICLR 2026)
>
> In the revision, we emphasize our contribution.
>
> [1] Multimodal LLM Guided Exploration and Active Mapping using Fisher Information. (ICCV 2025)
>
> [2] GLEAM: Learning Generalizable Exploration Policy for Active Mapping in Complex 3D Indoor Scenes. (ICCV 2025)
>
> **W2: Lack of Ablation on VLM Necessity.**
>
> While the VLM is inherently necessary since facilitating downstream human–agent interaction is one of our core contributions, we agree this is an important ablation. We conduct controlled ablations replacing VLM selection with heuristic baselines while keeping the rest of the pipeline unchanged:
>
> | Waypoint Selection | Cov. ↑ | AUC ↑ |
> |---|---|---|
> | Nearest-to-Centroid | 47.96% | 45.93% |
> | Max-Frontier-Distance | 52.85% | 50.92% |
> | **VLM-based (Ours)** | **75.31%** | **65.70%** |
>
> Results show our VLM-based method significantly outperforms heuristic baselines, confirming that the VLM is critical. The VLM's core advantage lies in *scene semantic understanding*: it identifies and selects waypoints at transition points that provide access to the largest unexplored areas. In contrast, heuristic methods lack semantic understanding, often selecting suboptimal waypoints and missing passages to unexplored areas.
>
> **Q2: Strong experimental assumptions.**
>
> Thank you for the valuable comment.
>
> **Textured scenes.** To address this concern, we add a comparison on **HM3D** against the recent VLM-based active mapping method [1]. As HM3D is used in their original paper, we directly report the results from their publication. For a fair comparison, we use the same GPT-4o as [1].
>
> | Methods | Completeness ↑ |
> |---|---|
> | VLM-based baseline [1] | 49.76% |
> | Ours | **55.64%** |
>
> Results demonstrate that our method generalizes to textured environments.
>
> **Pose uncertainty.** Following GLEAM, we add Gaussian noise N(0, σ²) (unit: meter) to the ground-truth pose:
>
> | σ²  | Cov. ↑ |
> |-----|--------|
> | 0   | 75.31% |
> | 0.1 | 73.25% |
> | 0.3 | 70.64% |
>
> Results demonstrate that our method is robust to pose noise.
>
> **Continuous action spaces.** Our method produces continuous 3D metric actions. The VLM selects among discrete waypoint IDs in image space, and each selected ID is mapped to a continuous 3D coordinate via 2D–3D correspondences. Local motion is then executed by an A\* planner, which outputs continuous trajectories.
>
> **Q3: Failure scenarios.**
>
> We add the failure analysis in the revision. **(1) Narrow passages:** Limited BEV resolution can cause frontier detection to miss small openings, preventing discovery of connected regions. **(2) VLM reasoning errors:** the VLM occasionally overlooks waypoints leading to unexplored areas.

---

> > ### Author Rebuttal · Reviewer_EG3D · 2026-04-05
> >
> > Thank you for your detailed rebuttal. However, after careful re-evaluation, my core concerns remain unresolved, and I am maintaining my recommendation. Your rebuttal argues that because your method outperforms GLEAM (a trained baseline) on coverage and AUC, it demonstrates a "critical" scientific contribution. This is a logical fallacy. A system achieving higher scores does not automatically imply a novel machine learning insight.
> >
> > 1. GLEAM is a learned policy. You are not comparing against a learned representation or a learned reasoning module.
> >
> > 2. Your "insight" is that a 235B parameter VLM, prompted with engineered text and projected waypoints, can pick a good candidate from a list of 25. Is it surprising that Qwen3-VL-235B outperforms a lightweight RL policy trained from scratch? No. That is an expected outcome given the sheer scale of priors in the VLM.
> >
> > 3. The central scientific question for ML venues is: What new principle of learning, representation, or reasoning does this reveal? Your answer is "representation and framework design are critical." That is a truism, not a discovery. You did not invent BEV, farthest-point sampling, or frontier-based exploration. You assembled them.

---

> > > ### Author Response · Authors · 2026-04-05
> > >
> > > We sincerely thank the reviewer for the continued discussion. We respectfully address each point below.
> > >
> > > **Q1. The contribution rests on performance numbers rather than genuine technical novelty.**
> > >
> > > We would like to gently clarify that our rebuttal argued: through well-designed representations and decision frameworks, VLMs can fully unleash their spatial reasoning to surpass task-specifically learned policies. Performance numbers serve as **evidence of effectiveness**, not as the contribution itself. Each component is independently validated by controlled ablations (Tables 4–5 and rebuttal experiments), where changing one component causes large performance shifts. We kindly ask the reviewer to consider the ablation evidence as a whole.
> > >
> > > **Q2. The baselines lack comparisons against learned representations or reasoning modules.**
> > >
> > > We follow the new GLEAM benchmark (ICCV 2025), which is the established large-scale complex active mapping benchmark. We respectfully note that the task itself does not mandate comparisons exclusively against learned representations or reasoning modules. In fact, the GLEAM benchmark itself includes non-learned baselines. Nevertheless, we did supplement a comparison against a VLM-based method [1] on HM3D using the same GPT-4o **in our rebuttal**, and our framework achieves clear advantages (55.64% vs. 49.76%).
> > >
> > > [1] Multimodal LLM Guided Exploration and Active Mapping using Fisher Information. (ICCV 2025)
> > >
> > > **Q3. A 235B VLM naturally outperforms an RL policy; the result is unsurprising.**
> > >
> > > We respectfully submit that this is **empirically false**. The same 235B VLM with a different action interface achieves only **45.10%** coverage (Table 4), which is **far below** GLEAM's 66.50%:
> > >
> > > | Setting (same Qwen3-VL-235B) | Cov. ↑ |
> > > |---|---|
> > > | Coordinate prediction | 45.10% |
> > > | **Ours** | **75.31%** |
> > >
> > > Without our framework, the 235B model **loses to** the RL policy (GLEAM). This directly shows that model scale alone is insufficient and that our novel framework design is the decisive factor.
> > >
> > > **Q4. You did not invent BEV, farthest-point sampling, or frontier-based exploration. You assembled them."**
> > >
> > > We respectfully disagree on two grounds.
> > >
> > > **First, we respectfully note that the reviewer's characterization does not accurately reflect our actual contributions.** We kindly ask the reviewer to re-examine our contributions, which were also recognized by other reviewers:
> > >
> > > - **360-BEV Representation:** A fused representation that combines omnidirectional semantics with BEV-aligned geometry to enable VLMs to jointly reason about semantics and geometry. Removing it causes coverage to collapse (75.31% → 38.77%, Table 4). This is not "using BEV" but rather designing a specific panoramic-semantic and BEV-geometric fusion tailored for VLM perception.
> > >
> > > - **Candidate Waypoint Generation:** A mechanism that generates modality-aligned candidate waypoints, aligns semantic-geometric cues, and directly converts VLM-selected image-space IDs into executable 3D actions. Replacing it with coordinate prediction drops coverage by 30 pp (Table 4). This design bridges the VLM's image-space reasoning to embodied 3D actions.
> > >
> > > - **Depth-First Exploration:** A VLM-driven agent that decomposes scenes into a topology tree with memory-based insertion and pruning, enabling efficient exploration with backtracking. This is a structured exploration strategy that leverages VLM scene understanding.
> > >
> > > - **Human-Agent Interaction (we respectfully note this contribution has been overlooked in the reviewer's assessment):** The scene topology tree provides a map prior for language-driven interaction: users specify target regions via coarse tree-node localization plus short descriptions, enabling concise map-updating or navigation instructions without complex scene descriptions (Fig. 7). This is a new capability absent from all prior active mapping methods.
> > >
> > > **Second.** Chain-of-Thought (NeurIPS 2022) did not invent prompting or LLMs; it identified a specific prompting strategy and empirically demonstrated its effectiveness. **Our work similarly identifies specific design principles for VLM-based active mapping in complex environments and validates each through controlled ablations.**
> > >
> > > We sincerely hope these points help address the reviewer's concerns. We are deeply grateful for the constructive comments and would be most thankful if the reviewer could kindly reconsider.

---

### Official Review · Reviewer_62or · 2026-03-10

**Soundness:** 3
**Presentation:** 3
**Significance:** 3
**Originality:** 3
**Overall Recommendation:** 3
**Confidence:** 4

**Summary:**

**Summary:**

This paper proposes a zero-shot active mapping method based on a visual language model. By fusing 360° panoramic images with BEV geometric representations, and combining a candidate waypoint generation strategy with a depth-first exploration strategy, it achieves active mapping in large-scale scenes without task-specific training and supports language-driven human-computer interaction. The overall work is complete, and the experimental results are impressive, but there are still shortcomings in terms of innovation and evaluation adequacy.

**Compliance With Llm Reviewing Policy:**

Affirmed.

**Final Justification:**

Thank the authors for their detailed response. The author's rebuttal resolved some of my concerns about the experiments, but after reading Reviewer EG3D's comments, I agree that the core contribution of this paper is too biased towards the engineering system integration of robots, so I have decided to maintain the original score.

**Key Questions For Authors:**

See weaknesses.

I will raise the score if the authors address my questions.

**Limitations:**

yes

**Strengths And Weaknesses:**

## Strengths

1. Without any task-specific training, the method outperforms the strongest supervised baseline in coverage and AUC by 13.25% and 14.00%, respectively, on GLEAM-Bench, fully demonstrating the potential of VLM-driven methods.

2. The approach of aligning omnidirectional semantics with BEV geometry effectively compensates for the limitations of a single representation, and ablation experiments clearly verify its key contribution to performance.

3. The design of selecting waypoints in image space and then backprojecting them to 3D space effectively solves the problem that VLM is not good at directly outputting metric actions, which is a practical and elegant engineering design.

4. The map update function based on scene topology trees and natural language commands has strong application significance in practical deployment scenarios.


### Weaknesses

1. Using VLMs for value point search and selection in image space is a common paradigm, and many works have adopted similar approaches. The core contribution of this paper lies more in the combinatorial innovation at the engineering level, rather than proposing fundamental new methods or insights. It is recommended that the authors more clearly explain the essential differences between this paper and existing VLM-driven navigation/mapping work.

2. The paper only compares two sizes of Qwen3-VL (8B vs 235B), lacking comparative experiments with other mainstream VLMs (such as GPT-4V, LLaVA, InternVL, etc.). It is currently impossible to determine whether the performance improvement of the method comes from the framework design itself or mainly depends on the capabilities of the selected ultra-large-scale model, which has a significant impact on the generalizability of the method.

3. Experiments are only evaluated on the single GLEAM-Bench benchmark, lacking validation on other mainstream scene exploration datasets (such as HM3D, Matterport3D, Replica, etc.). Furthermore, the baseline method lacks comparisons with other recent VLM-based exploration methods, making it difficult to comprehensively assess the true position of this method within the current technological level.

4. Note that the header of this paper displays "Submission and Formatting Instructions for ICML 2026." This is usually caused by the author directly using the official ICML 2026 LaTeX template without replacing the header with the paper title, which is a formatting issue.

---

> ### Author Rebuttal · Authors · 2026-03-31
>
> We sincerely thank the reviewer for the constructive feedback, including the recognition of our strong zero-shot performance, the effectiveness of the semantic-geometric alignment, the practical waypoint selection design, and the significance of the language-driven map-update functionality.
>
> **W1: It is recommended that the authors more clearly explain the essential differences between this paper and existing VLM-driven work.**
>
> Thank you for the constructive comment. To our knowledge, the most relevant prior work on VLM-based active mapping is **multimodal-active [1].”** Our method is distinguished by the following properties:
>
> **(1) Efficient active mapping in complex scenes.** Compared to multimodal-active, we focus on **efficient active mapping in large-scale complex scenes** under the GLEAM [2] setting. Specifically, we conduct experiments on the same HM3D dataset: our action planning takes about **2s per step**, while multimodal-active reports **more than 15s**, highlighting the efficiency and scalability of our design.
>
> **(2) Efficient 360-BEV representation.**
> multimodal-active prompts the VLM with a bird’s-eye rendering from a **growing global 3DGS map**, so planning cost increases with scene scale. Our method uses a lightweight **fused 360-BEV representation**, where the VLM reasons only over the **current 360 observation and incremental BEV**. This keeps the VLM planning cost **independent of explored scene size**. In addition, our representation also provides **omnidirectional semantic context and explicit geometric cues** for better decision-making.
>
> **(3) Modality-aligned decision-making.**
> multimodal-active use the VLM to select a **coarse frontier region**, and then derive the final 3D action via Fisher-information computation over the 3D scene. In contrast, we generate **fine-grained candidate waypoints** with aligned **semantic and geometric cues** in the 360-BEV representation, and directly convert the VLM-selected waypoint into an **executable 3D action**. This design reduces computation while enabling **finer-grained semantic decision-making** by the VLM.
>
> **(4) Efficient exploration through local decisions with global memory.**
> Instead of reasoning over the **entire explored map** and selecting only one target at a time, our VLM reasons over local observations and selects **all informative waypoints per step**, which are maintained in a **scene topology tree** with memory for insertion and pruning. This allows us to make fuller use of each VLM inference and **perform backtracking** without repeatedly replanning, reducing action-planning cost.
>
> **(5) Scene-tree prior for interaction.**
> The **scene topology tree** also provides a useful prior for **human–agent interaction**. Users can specify a target region via **coarse tree-node localization plus a short language description**, enabling concise map-updating or navigation instructions without requiring complex scene descriptions.
>
> In the revision, we emphasize our contribution.
>
> **W2: Missing comparison with other mainstream VLMs.**
>
> Thank you for this important suggestion. To address this concern, we add experiments with multiple VLM models while keeping the rest of the pipeline unchanged.
>
> | VLM Backbone | Cov. ↑ | AUC ↑ | CD ↓ |
> |---|---:|---:|---:|
> | Gemini-2.5-flash | 71.73% | 63.07% | 0.95m |
> |GPT-4o | 72.47% | 63.85% | 0.83m |
> | Qwen3-VL-235B-A22B (ours) | 75.31% | 65.70% | 0.75m |
>
> Results show that our framework consistently remains competitive across diverse VLM families and outperforms the **baseline GLEAM (66.50% Cov.)**, suggesting that the gains mainly come from the framework design rather than dependence on a specific ultra-large backbone. We include these results in the revised paper.
>
> **W3: Limited evaluation.**
>
> Thank you for this valuable suggestion. To address this concern, we add a comparison on **HM3D** against the recent VLM-based active mapping method [1]. As HM3D is the benchmark used in their original paper, we directly report the results from their publication. For a fair comparison, we use the same GPT-4o as [1].
>
> | Methods | Completeness ↑ |
> |---|---|
> | VLM-based baseline [1] | 49.76% |
> | Ours | **55.64%** |
>
> As shown in the table, our method outperforms this recent VLM-based active mapping method on HM3D, demonstrating that our framework exhibits strong **zero-shot transfer** capability to another mainstream scene exploration benchmark. We included these comparisons in the revised paper.
>
> **W4: Formatting error — ICML 2026 header.**
>
> We sincerely apologize for this formatting oversight. We have corrected this in the revised manuscript and conducted a thorough check to ensure no other template-default artifacts remain.
>
> [1] Multimodal LLM Guided Exploration and Active Mapping using Fisher Information. (ICCV 2025)
>
> [2] GLEAM: Learning Generalizable Exploration Policy for Active Mapping  in Complex 3D Indoor Scenes. (ICCV 2025)

---

> > ### Author Rebuttal · Reviewer_62or · 2026-04-02
> >
> > Thank the authors for their detailed response. The author's rebuttal resolved some of my concerns about the experiments, but after reading Reviewer EG3D's comments, I agree that the core contribution of this paper is too biased towards the engineering system integration of robots, so I have decided to maintain the original score.

---

> > > ### Author Response · Authors · 2026-04-02
> > >
> > > We sincerely thank the reviewer for the time and valuable feedback. We genuinely hope the following clarification can help demonstrate the value of our contributions, and we would be grateful if the reviewer could kindly reconsider.
> > >
> > > **(1) Representation and decision framework design are critical contributions.**
> > >
> > > Our zero-shot method outperforms GLEAM (trained with RL on large-scale task-specific data), with relative improvements of approximately **13.25% in coverage and 14.00% in AUC**. This demonstrates that through well-designed representations and decision frameworks, VLMs can fully unleash their spatial reasoning to surpass task-specifically learned policies. This carries significant implications for the ML community: **beyond model architectures and training, representation and framework design are equally critical**. We would like to kindly highlight that each of our components addresses a fundamental question:
> > >
> > > - **360-BEV Representation:** *What representation best enables VLMs to jointly reason about semantics and geometry?* Table 4 shows removing it causes coverage to drop drastically (75.31% → 38.77%), confirming representation design is the critical factor.
> > >
> > > - **Candidate Waypoint Generation:** *How to effectively unlock VLMs' capabilities for active mapping?* A waypoint generation mechanism that aligns semantic–geometric cues and directly converts VLM-selected waypoints into executable 3D actions, enabling fine-grained decision-making with efficient action planning.
> > >
> > > - **Depth-First Exploration:** *How to achieve thorough coverage in unseen large-scale scenes?* A scene topology tree with memory-based insertion and pruning, supporting efficient depth-first exploration with backtracking.
> > >
> > > - **Human–Agent Interaction:** *How to enable concise language-driven interaction?* The scene topology tree provides a map prior: users specify target regions via coarse tree-node localization plus short descriptions, enabling concise map-updating or navigation instructions without requiring complex scene descriptions (Fig. 7).
> > >
> > > **(2) This scope aligns with the themes of top ML venues.**
> > >
> > > We respectfully note that top ML venues consistently accept works that design novel frameworks to unlock VLM/LLM capabilities without additional training:
> > >
> > > - *Chain-of-Thought Prompting Elicits Reasoning in Large Language Models* (NeurIPS 2022)
> > > - *LLMs Can See and Hear Without Any Training* (ICML 2025)
> > > - *A Training-Free Framework for Long Video Understanding via Video-Query-Options Similarity* (ICLR 2026)
> > >
> > > These works demonstrate that principled framework and representation design, even without new model training, constitutes a valuable ML contribution. Our work follows this spirit by designing representations and decision frameworks that unlock VLM spatial reasoning for embodied tasks.
> > >
> > > We sincerely hope the reviewer might consider revisiting the score in light of these clarifications, and we truly appreciate the constructive feedback that has helped us strengthen the paper.

---

### Official Review · Reviewer_5LPh · 2026-03-11

**Soundness:** 4
**Presentation:** 4
**Significance:** 3
**Originality:** 4
**Overall Recommendation:** 5
**Confidence:** 3

**Summary:**

This paper presents a zero-shot active mapping framework based on a VLM for large-scale unseen environments. The key idea is to avoid asking the VLM to output precise 3D actions directly; instead, the method builds a fused 360 image + BEV occupancy representation, generates geometrically valid candidate waypoints in BEV, projects them into the 360 image, and lets the VLM select waypoint IDs in image space, which are then mapped back to executable 3D actions. On top of this, the system maintains a topology tree and performs depth-first exploration to improve systematic coverage, while also supporting language-guided human-agent interaction for targeted map updating. Experiments on GLEAM-Bench show clear gains over prior baselines in coverage and exploration efficiency both quantitatively and qualitatively. human-agent interaction results are also showed.

**Compliance With Llm Reviewing Policy:**

Affirmed.

**Key Questions For Authors:**

See the weakness

**Limitations:**

yes

**Strengths And Weaknesses:**

Strengths

1. Important problem setting: Zero-shot exploration is an important capability for embodied agents operating in unseen environments.

2. Novel method design: The paper leverages a VLM to reason over 2D panoramic images to select candidate exploration waypoints, and maintains a topological tree structure to organize explored regions and perform depth-first exploration.

3. Good experimental validation: The paper provides both quantitative and qualitative results, demonstrating clear improvements over prior methods. It also showcases human–agent interaction scenarios, illustrating the flexibility of the proposed framework.

Weakness

It would be valuable to include real-world experiments, even without comparisons to other methods, to demonstrate that the approach works beyond simulation.

---

> ### Author Rebuttal · Authors · 2026-03-31
>
> We sincerely thank the reviewer for the positive assessment of our work, including the recognition of the important problem setting, the novel method design, the strong experimental validation, and the human-agent interaction capability.
>
> **W: It would be valuable to include real-world experiments, even without comparisons to other methods, to demonstrate that the approach works beyond simulation.**
>
> We agree that the real-world experiment is valuable. We address this from three perspectives:
>
> **1. Existing evidence from real-scan scenes.** We would like to kindly note that GLEAM-Bench already includes **real-scan scenes** reconstructed from real-world environments (Table 1). On these challenging scenes, our zero-shot method achieves **57.90%** coverage, surpassing the strongest trained baseline GLEAM (**47.04%**). We believe this provides partial evidence of generalization to real-world scene geometry and complexity.
>
> **2. Structurally narrower sim-to-real gap.** We would also like to highlight that our method has a fundamentally narrower sim-to-real gap than learned baselines, due to its architectural design:
>
> - **(a) Zero-shot VLM with real-world priors.** The VLM used in our method is pre-trained on large-scale real-world data and requires **no task-specific fine-tuning**. Its scene understanding and spatial reasoning capabilities are expected to transfer naturally to real environments.
>
> - **(b) No simulation-trained components.** Unlike learning-based baselines, which train models and policies in simulation and are therefore more vulnerable to domain shift, our method does not rely on simulator-trained perception or policy networks, removing a major source of sim-to-real degradation.
>
> **3. Real-world deployment.** Following the valuable suggestion, we deploy our method on a **TurtleBot4 Lite** equipped with an **Insta360** camera (360° RGB) and an onboard **OAK-D** camera (depth). The robot communicates with a remote workstation via **ROS 2**: panoramic images, depth measurements, and IMU readings are streamed to the server for VLM-based perception and planning. The multimodal reasoning module is implemented by accessing **Qwen3-VL-235B-A22B-Instruct** through an API. The resulting navigation commands are then sent back to the robot for execution. We plan to add a visualization video and qualitative results in the revised paper.

---

> > ### Author Rebuttal · Reviewer_5LPh · 2026-04-04
> >
> > the writers give full results to my concerns and weakness

---

> > > ### Author Response · Authors · 2026-04-05
> > >
> > > We sincerely thank the reviewer for the time and constructive feedback throughout the review process. We are grateful that all concerns have been fully resolved, and we once again appreciate the reviewer's positive assessment of our work. All discussed improvements have been incorporated into the revised manuscript.

---

### Decision · Program_Chairs · 2026-04-30

**Decision:**

Accept (regular)

**Comment:**

After discussion and consideration of the rebuttal, I recommend weak accept. Reviewers agree that the paper addresses an important problem in zero-shot active mapping and that the proposed framework is technically solid, with strong empirical gains over prior baselines on the benchmark considered. In particular, the combination of the fused 360-BEV representation, the waypoint selection mechanism that aligns VLM reasoning with executable actions, and the exploration strategy appears effective, and the rebuttal usefully strengthened the paper through additional ablations and comparisons addressing several reviewer concerns. At the same time, I find the concerns about the level of conceptual novelty for ICML to be legitimate: the main contribution is more in the design and integration of representations and decision-making modules around an off-the-shelf VLM than in a fundamentally new learning method. Still, on balance, I believe the empirical improvements, careful system design, and practical relevance make this a worthwhile contribution, though it sits closer to the acceptance threshold than a clear accept.